# FACTORSIM: Generative Simulation via Factorized Representation

**Fan-Yun Sun**
Stanford University

**S. I. Harini**
Stanford University

**Angela Yi**
Stanford University

**Yihan Zhou**
Stanford University

**Alex Zook**
Nvidia

**Jonathan Tremblay**
Nvidia

**Logan Cross**
Stanford University

**Jiajun Wu**
Stanford University

**Nick Haber**
Stanford University

## Abstract

Generating simulations to train intelligent agents in game-playing and robotics from natural language input, from user input or task documentation, remains an open-ended challenge. Existing approaches focus on parts of this challenge, such as generating reward functions or task hyperparameters. Unlike previous work, we introduce FACTORSIM that generates full simulations in code from language input that can be used to train agents. Exploiting the structural modularity specific to coded simulations, we propose to use a **factored** partially observable Markov decision process representation that allows us to reduce context dependence during each step of the generation. For evaluation, we introduce a *generative simulation* benchmark that assesses the generated simulation code's accuracy and effectiveness in facilitating zero-shot transfers in reinforcement learning settings. We show that FACTORSIM outperforms existing methods in generating simulations regarding prompt alignment (*i.e.*, accuracy), zero-shot transfer abilities, and human evaluation. We also demonstrate its effectiveness in generating robotic tasks.

## 1   Introduction

Simulations hold significant potential for training agents to perform real-world tasks where data collection is costly, dangerous, or infringes on individual privacy. A major bottleneck in harnessing the potential of simulations at scale for agent training is the cost of designing and developing them, especially when we need a distribution of simulations that meet detailed design specifications to train more generalized policies. In this paper, we aim to generate coded simulations given text specifications. Code provides a natural interface for users to inspect, modify, and debug the simulation. It also allows us to craft diverse environments for Reinforcement Learning (RL) purposes.

Generating full simulations in code to train agents from a text prompt is an under-explored challenge. Previous works focus on parts of this challenge, including reward function design [22], hyperparameter tuning [24], and task configuration while relying on an existing simulator [35]. These methods use large language models (LLMs) to generate the components of simulations specified as code. However, when faced with large and detailed contexts, LLMs often generate simulations that ignore or fail to adhere to parts of the input prompt [21]. This issue is not solely due to the limitations of

---

[1] Work done while Harini S I was an intern at Stanford.

[2] Correspondence to sunfanyun@cs.stanford.edu

[3] Project website: https://cs.stanford.edu/ sunfanyun/factorsim/

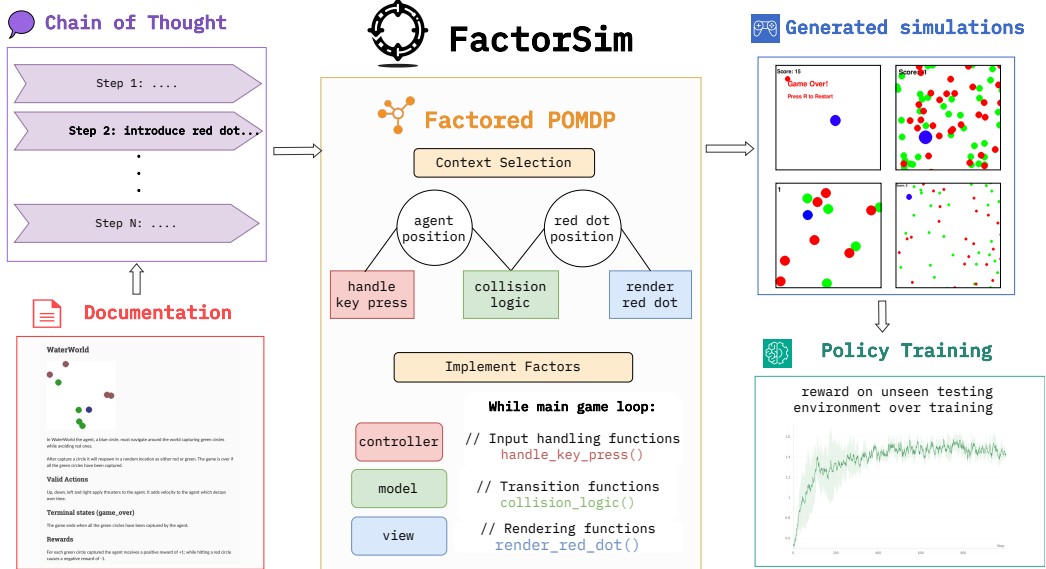

Figure 1: Overview of FACTORSIM. FACTORSIM takes language documentation as input, uses Chain-of-Thought to derive a series of steps to be implemented, adopts a Factored POMDP representation to facilitate efficient context selection during each generation step, trains agents on the generated simulations, and tests the resulting policy on previously unseen RL environments.

existing LLMs but also suggests that some form of decomposition is always critical as we scale up the number of components in simulations. We ask the question: can we exploit the inherent structure (e.g., having a game loop that handles agent actions, updates internal game states accordingly, and displays the game states to the users through a rendering process) of coded simulations to generate them better?

We propose FACTORSIM, a framework that takes an arbitrary language specification as input and outputs a full simulation that can be used to train RL agents. The key idea of FACTORSIM is to decompose the input prompt into a series of steps and then use a factored Partially Observable Markov Decision Process (POMDP) representation to reduce the context needed for each generation step. To realize FACTORSIM, we use the *model-view-controller* software design pattern to structure the generation process. Consider generating a coded simulation of WaterWorld; see Figure 1. The game consists of an agent (blue circle) traveling in a 2d world, capturing food (green circle) while avoiding enemies (red circle). Our method first decomposes the game description into multiple steps to be implemented. For example, a step instruction could be "Introduce red dot enemies that can be controlled with arrow keys. Give the player a -1 reward when the agent collides with an enemy". We first select the context needed for this functionality to be implemented, *e.g.*, positions of existing agents. Subsequently, FACTORSIM generates (at most) three functions: one to handle player input (i.e., *handle_key_press*, the controller component), one to implement the collision logic (i.e., *collision_logic*, the model component), and one to update the rendering function (i.e., *render_red_dot*, the view component). Limiting the context during each step of the simulation generation process allows FACTORSIM to focus on the task at hand while avoiding hallucinating non-existent functions or modifying code not meant to be changed.

To evaluate the task of full simulation generation, we propose a new *Generative Simulation*[4] benchmark with accompanying success metrics. One set of success metrics is the pass rate in automated system tests. Commonly used in game development, these system tests programmatically assess whether the behavior of the generated simulation adheres to the specifications given in the input prompt. The second success metric assesses the value of the generated simulations for transfer learning in an RL setting. This evaluates how well agents trained on a set of generated simulations can generalize to held-out environments that satisfy the design specifications provided in prompts. Generalization to unseen environments is crucial for many applications, including transferring robotics

---

[4]We adopt this term from [39] to refer to automated simulation generation to train agents within.

policies learned in simulation to the real world. This benchmark consists of 8 RL environments with varying levels of difficulty. In addition to evaluating our method on the benchmark we introduced, we further assess FACTORSIM's ability to generate robotic tasks on the dataset published by Gen-Sim [35]. We demonstrate the value of our method, FACTORSIM, on both the benchmark task suite and GenSim's dataset, showing performance superior to baseline alternatives.

In summary, our contributions are three-fold. First, we propose FACTORSIM, a framework for generating coded simulation with a factor graph of a POMDP as a principled way to reduce context dependence. Second, we introduce a new generative simulation benchmark by adapting an existing RL benchmark [33], and demonstrate FACTORSIM's superior results against baselines in terms of code correctness (i.e., prompt alignment), ability to facilitate zero-shot generalization and human evaluation of the simulations. Third, we demonstrate that FACTORSIM can be applied to generating simulation tasks for robotics, outperforming existing approaches.

## 2 Related Work

We aim to generate simulations for training agents to generalize to previously unseen environments. Recent work has investigated this in the context of learned neural world models and LLM-generated code for simulations.

World models simulate the dynamics of a given environment and use this as a proxy environment for agent training, rather than interacting with a ground truth simulator [8]. Several approaches have demonstrated the value of learning world models as part of general algorithms that can learn to play a variety of games (AlphaZero [30], Muesli [13], and DreamerV3 [9]). Other efforts use a large set of offline data to learn a world model that is subsequently used for agent training, including for autonomous vehicle driving (GAIA-1 [14]), robotic manipulation (UniSim [40]), and 2D platformer games (Genie [5]). We generate world models as code as they are more interpretable, modular, and easily modified or extended by humans—key advantages we believe are important for their use in authoring large-scale or complex simulations.

LLMs have generated many parts of simulations for game playing and robotics. In (RL) games, LLMs have been used to generate game levels [34, 31], to choose parameters for an existing simulator [44], and to assist humans in creating full games [3]. In robotics, LLMs have been used to generate reward functions, task specifications, and specific components like scene configurations within robotics tasks. Many works such as RoboGen [37], Holodeck [41], and Gen2Sim [16] build on top of existing simulators and use a series of prompts to generate interactable 3D environments to train agents. GenSim [35] starts from a human task library and iteratively generates and tests new tasks to generate robotic manipulation tasks. Other efforts have focused on generating reward functions for tasks [22, 19, 23]. Eureka [22] uses feedback from agent training to refine reward function specification. Our approach is able to generate both the simulator dynamics and reward functions and can be applied to both robotics and games.

As noted above, LLMs can struggle to handle complex tasks: this has prompted research into different ways to structure LLM reasoning. Chain-of-Thought (CoT) prompting demonstrated LLM performance can be substantially boosted by prompting the LLM to break a single task into multiple steps with either few-shot examples [38] or zero-shot [18]. Subsequent work has developed a variety of techniques to improve LLM reasoning through multi-step reasoning prompts: checking for consistency among multiple reasoning paths [36], interleaving reasoning and tool use (ReACT [43]), using tree data structures to guide the LLM reasoning process (Tree-of-Thought [42]), or formulating reasoning as a tree search process [12, 46]. Approaches for general code generation include decomposing the task into functions to subsequently generate (Parsel [45]), generating code to reach a series of intermediate execution states (ExeDec [28]), and using a multi-agent framework to generate, test, and refine code (AgentCoder [15]). Other efforts optimize the prompts for given tasks, using evolutionary search (EvoPrompt [7]) or defining generalized declarative programming frameworks with modular optimization algorithms [17]. Our approach generates code by leveraging a factorized representation specific to simulations to reduce the input context needed for different reasoning steps; it can be used in conjunction with approaches for general code generation, such as generating tests as a form of self verification.

# 3 FACTORSIM: Generating Simulations via Factorized Representation

A simulation is a structured system of modules connected by events and responses. Our framework, FACTORSIM, generates code using LLMs by exploiting this structure to construct a simulation progressively. Our key insight is that, by generating a simulation step-by-step while **only selecting the relevant context information needed for each step**, we can effectively reduce the reasoning capacity needed for each step, leading to simulations that adhere more closely to the input requirements.

In this section, we describe our method for generating Turing-computable simulations. First, we describe simulations that can be modeled as a Partially Observable Markov Decision Process (POMDP). Second, we use Chain-of-Thought (CoT) to decompose an input prompt describing the desired full simulation into a series of prompts describing different components to be implemented. Third, we introduce a factorized POMDP representation that exploits the inherent modularity of coded simulations. Refer to Algorithm 1 and Figure for an overview of FACTORSIM alongside an illustrative example.

---

**Algorithm 1:** FACTORSIM

---

**Input:** $Q_{\text{text}}$, a natural language description of the simulation, and an LLM
**Output:** a turing-computable simulation represented as a POMDP $\mathcal{M}' = \langle S, A, O, T, \Omega, R \rangle$

Initialize a Factored POMDP $\mathcal{M}_1 \leftarrow \langle S_1, A, \emptyset, T_1, \emptyset, R_1 \rangle$ where
- $S_1 := \{s_{\text{score}}\}$
- $A$ is the set of all keyboard inputs
- $T_1$ is an identity function, i.e., $T_1(s' \mid s, a) = \mathbf{1}[s' = s]$
- $R_1(s, a, s') := s'_{\text{score}} - s_{\text{score}}$

// Chain of Thought
Derive a step-by-step plan $(q_1, \ldots, q_k) \sim p(q_1, \ldots, q_k \mid Q_{\text{text}})$                    Eq. (1)

**for** *each step, or module $q_k$* **do**

    **State space update & context selection** $p(S_{k+1}, S[Z_k] \mid S_k, q_k)$          Eq. (9),(10)

    // Controller component update
    **Action-dependent state transition model update**: $p(T_{k+1}^{(a)} \mid S[Z_k], A, q_k)$

    // Model component update
    **Action-independent state transition model update**: $p(T_{k+1}^{(s)} \mid T[Z_k], S[Z_k], q_k)$

    // View component update
    **Observation model update**: $p(\Omega_{k+1} \mid S[Z_k], q_k)$          Eq. (13)

    $\mathcal{M}_{k+1} = \langle S_{k+1}, A, O_{k+1}, T_{k+1}, \Omega_{k+1}, R_1 \rangle$ where $O_{k+1}$ is the new observation space defined by $S_{k+1}$ and $\Omega_{k+1}$, and $T_{k+1}(s' \mid s, a) = T_{k+1}^{(s)}(s' \mid s) \cdot T_{k+1}^{(a)}(s \mid s, a)$.

**end**
Return the final simulation $\mathcal{M}' \leftarrow \mathcal{M}_{k+1}$

---

## 3.1 Modeling Simulation as POMDP

A Partially Observable Markov Decision Process (POMDP) is used to represent a coded simulation. Formally a POMDP is represented as a tuple $\mathcal{M} = \langle S, A, O, T, \Omega, R \rangle$ where $S$ is a set of states, $A$ is a set of actions, $O$ is a set of observations, $T : S \times A \to \mathbf{\Delta}(S)$ is a transition probability distribution, $\Omega : S \to \mathbf{\Delta}(O)$ is an observation function, and $R : S \times A \times S' \to \mathbb{R}$ is the reward model [5].

We aim to generate a simulation from a prompt $Q_{\text{text}}$. In this paper, we are particularly interested in the case where $Q_{\text{text}}$ comprises detailed design specifications such that the resulting simulation could be used to train agents, though our method applies to any prompt for defining a simulation. In our experiments, $Q_{\text{text}}$ is a paragraph of text around 10 sentences specifying this simulation.

---

[5]We omit the discount factor $\gamma$ and the initial state distribution $\pi$ in the formulation for brevity. In our experiments, $\pi$ is generated alongside the states $S$.

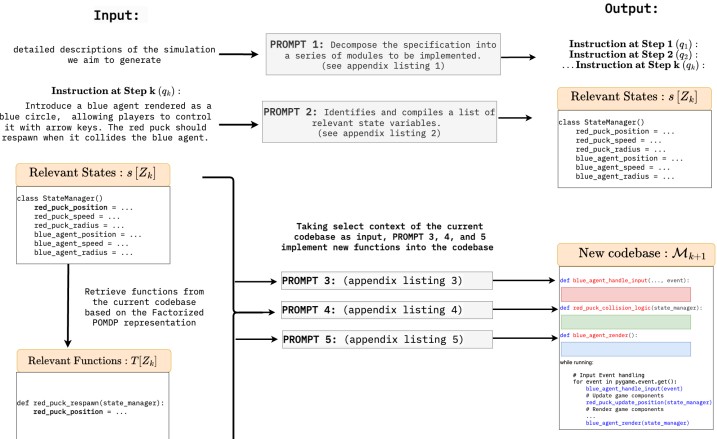

Figure 2: An illustrative example of how the five main prompts in FactorSim correspond to our formulation in Algorithm 1. Note that the function *red_puck_respawn* is retrieved as part of the context to Prompt 3, 4, and 5 because it modifies the state variable *red_puck_position*, a state variable LLM identified as relevant in prompt 2.

## 3.2 Chain of Thought

We first decompose the prompt $Q_{\text{text}}$ into a series of steps using Chain of Thought [38], each describing a module of the simulation to be implemented. Following similar formulation as in [26], this can be thought of as marginalizing over a step-by-step plan variable $(q_1, \ldots, q_k)$ using $N$ Monte Carlo samples:

$$\hat{p}(\mathcal{M}'|Q_{\text{text}}) = \frac{1}{N} \sum_{i=1}^{N} p(\mathcal{M}'|q_1^{(i)}, \ldots, q_K^{(i)}), \quad \text{where } (q_1^{(i)}, \ldots, q_K^{(i)}) \sim p(q_1, \ldots, q_K|Q_{text}), \quad (1)$$

$p$ is a probability estimation model (i.e., an LLM in our experiments), and $\mathcal{M}'$ is the resulting code that fully specifies a simulation. In practice, we only produce a single plan $N = 1$.

Intuitively, this process breaks the prompt into sub-tasks. After we sample such a plan of $K$ steps, we generate the simulation progressively. Given an existing POMDP $\mathcal{M}$ and a natural language specification $q$, we update the POMDP to reflect the changes specified.

$$p(\mathcal{M}_{K+1}|q_1, \ldots, q_K) \approx \prod_{k=1}^{K} p(\mathcal{M}_{k+1}|\mathcal{M}_k, q_k) \quad (2)$$

where $\mathcal{M}_{k+1}$ is the POMDP (simulation as code) after the $k$-th step is implemented, and $\mathcal{M}_{K+1}$ is the final simulation. While Chain-of-Thought prompting allows LLMs to avoid having to generate code for all simulation logic at once, the complexity of each step still grows with $k$ due to the expanding codebase. This task remains challenging because LLMs must comprehend the code and accurately identify where modifications are needed. Acknowledging the limited reasoning ability of LLMs, we ask: can we further decompose the $p(\mathcal{M}_{k+1}|\mathcal{M}_k, q_k)$ into simpler distributions to reduce the complexity of each prompt?

## 3.3 Decomposition by Factorized Representation

Naively, we could further decompose a step of the generation into several steps, each focused on generating a different component of the POMDP:

$$p(\mathcal{M}_{k+1}|\mathcal{M}_k, q_k) = p(S_{k+1}|\mathcal{M}_k, q_k) \cdot \quad (3)$$
$$p(T_{k+1}|S_{k+1}, \mathcal{M}_k, q_k) \cdot \quad (4)$$
$$p(R_{k+1}|S_{k+1}, T_{k+1}, \mathcal{M}_k, q_k) \cdot \quad (5)$$
$$p(\Omega_{k+1}|S_{k+1}, T_{k+1}, R_{k+1}, \mathcal{M}_k, q_k) \quad (6)$$

However, this still requires the LLMs to take the entire simulation ($\mathcal{M}_k$) as context, which could be over hundreds of lines of code in our experiments. Empirically, we observe that many failed generations can be attributed to LLMs attending to or modifying parts of the input context unrelated to the prompt.

To reduce the input context needed for each generation step, we propose to use a factored POMDP representation to remove the dependence on the full previous POMDP as context. For instance, given an existing simulation $M_k$ of red, green, and blue agents, to implement the $kth$-step instruction $q_k$: `respawn the red agent when it collides with the blue agent`, we only need context regarding the respawn logic of the red agent and the positions of the red and blue agents. Code regarding the green agent or the rendering logic would be unnecessary context.

To formalize our approach, we first introduce notation common to the literature [25, 32]. Suppose we have a POMDP with a state space factored into $n$ state variables $S = S[1] \times \ldots S[n]$ and $Z$ is a subset of indices $Z \subseteq \{1, 2, \ldots, n\}$, we define the scope set $S[Z] := \bigotimes_{i \in Z} S[i]$ as the state space spanned by the subset of state variables. For example, if $Z = 1, 3, 4$, then $S[Z]$ defines a state space defined by $S[1] \times S[3] \times S[4]$. We denote a state in the scoped state space $S[Z]$ as $s[Z]$. Below, let us formally define a factored POMDP.

**Definition 3.1.** A factored POMDP is a POMDP with both factored transition distribution and factored reward function. A transition probability distribution $T$ of a POMDP with discrete action space is factored over its state space $S = S_1 \times \ldots S_n$ with scopes $Z_1, \ldots, Z_m$ if, for all $s \in \mathcal{S}, a \in A$ there exist some $\{T_i\}_{i=1}^m$ in the space of all possible transition distributions on the state space $S$ and action space $A$, such that,

$$T(s|s, a) = \prod_{i=1}^m T_i \left( s[i] \mid s[Z_i], a \right). \tag{7}$$

A reward function $R$ of a POMDP is factored over $S = S_1 \times \ldots S_n$ with scopes $Z_1, \ldots, Z_l$ if, for all $s \in \mathcal{S}, a \in A$ there exist some $\{R_i\}_{i=1}^l$ in the space of all possible reward functions on the state space $S$ and action space $A$, such that,

$$R(s, a) = \sum_{i=1}^l R_i \left( s[Z_i], a \right). \tag{8}$$

A factored POMDP can be represented as a factor graph [6] with two types of nodes: *state variables* (i.e., $S_i$) and *factors* (i.e., $T_i$ or $R_i$), functions of (state) variables. Our idea is to **reduce context dependence by structuring the code using a factored POMDP representation** and treat each generation step as expanding a factored POMDP with new state variables and new *factors*. During every step $q_k$, we first select a set of relevant state variable indices $Z_k$. Then, we select existing factors that have overlapping scope with the selected set of state variables as context, which we denote as $T[Z_k]$ and $R[Z_k]$. That is, we can reduce the dependence on the previous simulation $M_k$ and rewrite Equation 3-6 to the following:

$$
\begin{aligned}
p(\mathcal{M}_{k+1}|\mathcal{M}_k, q_k) \approx & p(S_{k+1}|S_k, q_k) \cdot && \text{update state space} && (9)\\
& p(S[Z_k]|S_{k+1}, q_k) \cdot && \text{identify relevant state variables} && (10)\\
& p(T_{k+1}|T[Z_k], S[Z_k], A, q_k) \cdot && \text{update state transition function} && (11)\\
& p(R_{k+1}|R[Z_k], S[Z_k], A, q_k) \cdot && \text{update reward function} && (12)\\
& p(\Omega_{k+1}|S[Z_k], q_k). && \text{update partial observation function} && (13)
\end{aligned}
$$

Note that $Z_k$ can only consist of state variable indices in the state space $S_{k+1}$. In practice, we achieve this by encouraging the LLM to select a minimal set of relevant states $Z_k$ in the prompt.

We find that the term 11 is most prone to error, likely because the most complicated functions of a simulation are state transitions. Motivated by this observation, we propose to adopt the *model-view-controller* design pattern for structuring these prompts. Instead of prompting LLMs to update the state transition function first and then update the reward function, we prompt the LLMs to update the action-dependent part of the state transition function (i.e. the *Controller* component) and then the

---

[6]More precisely, a factor graph of a Dynamic Bayesian Network (DBN) [11, 4].

Table 1: Percentage of system tests passed by different methods of generating 2D RL games.

| % of system tests passed. | Flappy Bird | Catcher | Snake | Pixelcopter | Pong | Puckworld | Waterworld | Monster Kong |
|---|---|---|---|---|---|---|---|---|
| Mistral-7B-Instruct | 0.00 | 0.00 | 0.00 | 0.00 | 0.00 | 0.00 | 0.00 | 0.00 |
| Llama-3 | 0.15 | 0.33 | 0.19 | 0.14 | 0.01 | 0.43 | 0.25 | 0.29 |
| Llama-3 w/ self debug | 0.15 | 0.41 | 0.28 | 0.19 | 0.03 | 0.44 | 0.22 | 0.31 |
| Llama-3 CoT w/ self debug | 0.20 | 0.39 | 0.25 | 0.21 | 0.16 | 0.50 | 0.42 | 0.35 |
| GPT-3.5 | 0.19 | 0.39 | 0.37 | 0.38 | 0.22 | 0.33 | 0.34 | 0.19 |
| GPT-4 | 0.35 | 0.35 | 0.42 | 0.44 | 0.25 | 0.34 | 0.46 | 0.21 |
| GPT-4 w/ self debug | 0.33 | 0.53 | 0.43 | 0.51 | **0.75** | 0.41 | 0.45 | 0.31 |
| GPT-4 w/ AgentCoder | 0.18 | 0.45 | 0.27 | 0.43 | 0.43 | 0.33 | 0.20 | 0.23 |
| GPT-4 CoT w/ self debug | 0.30 | 0.51 | 0.39 | 0.53 | 0.64 | 0.47 | 0.50 | 0.34 |
| Llama-3 w/ FACTORSIM (ours) | 0.55 | 0.54 | **0.50** | 0.41 | 0.38 | 0.58 | 0.27 | 0.35 |
| GPT-4 w/ FACTORSIM (ours) | **0.78** | **0.66** | 0.44 | **0.78** | 0.61 | **0.81** | **0.62** | **0.44** |

action-independent part (i.e., *Model*). We treat the reward model as part of the state transition function that updates a *score* state variable. That is, $T(s'|s, a) = T^{(s)}(s'|s)T^{(a)}(s|s, a)$ where $T^{(a)}(s|s, a)$ denotes the part of the state transition function that handles how actions affect the states and $T^{(s)}(s'|s)$ denotes the part of the state transition function that how states are updated every step. This gives us our final algorithm as illustrated in Algorithm 1.

In Algorithm 1, colors indicate the corresponding components of the model-view-controller pattern. Red highlights the *controller*, corresponding to parts of the state transition dependent on user-input actions.Green shows the *model*, corresponding to parts of the state transition function that are not dependent on user-input actions. Blue shows the *view* component, updating the observation function that acts as the "renderer" of the state space.

## 4  Experiments

In this paper, we consider two types of simulations: 2D Reinforcement Learning (RL) games and robotics tasks in a physics engine. We also introduce a new benchmark to evaluate generative simulation methods. Our experiments are designed to test three hypotheses. First, FACTORSIM generates simulations with *better prompt alignment*, which we evaluate through system tests and human evaluations. Second, FACTORSIM enables *better zero-shot transfer* by training RL agents in the simulated generated environments. Third, FACTORSIM's strengths in *generating robotic tasks*.

### 4.1  RL Game Generation

To answer our first two hypotheses, we propose a new benchmark that includes all 2D games from the PyGame Learning Environment [7] [33]: Flappy Bird, Catcher, Puckworld, Pixelcopter, Pong, Snake, Waterworld, and Monster Kong. For each RL game, the input prompt consists of the game's online documentation. Since most game documentation is incomplete, we manually supplement them with additional details (see Appendix). This ensures that our method and the baselines do not hallucinate any missing game information, allowing for a fair evaluation across all methods.

Following common practices in game development, we design system tests to verify that the generated simulations follow the specified logic programmatically. These tests simulate actions like key presses and mouse clicks and check if the game states are updated correctly. Refer to the Appendix for more details.

**Baselines**  For baselines, we compare to three methods using a closed-source (GPT-4 [1]) and an open-source LLM (Llama-3 [2]). The first approach prompts the LLM with all contexts at once, which we denote as the *vanilla* method. The second approach uses *self-debugging* [6], where the model retries generating the code when provided error messages from running the code (up to 10 times in our experiments). A third approach combines *Chain-of-Thought [38] (CoT)* reasoning with self-debugging, where the LLM generates code incrementally, processing one instruction at a time. Additionally, we incorporate AgentCoder [15] as a baseline. CoT with self-debugging is an ablation study of our method that acts without the factored POMDP representation.

---

[7]We exclude the sole 3D game Raycast Maze and leave 3D game generation to future work.

**Code Generation Evaluation**
Table 1 shows the results for the baselines and our method. FACTORSIM outperforms all baselines in 7 out of 8 games. Additionally, we compare performance and LLM token usage across various methods using GPT-4 (Figure 3). While the vanilla baseline uses the fewest tokens, it only achieves moderate accuracy. Additionally, combining Chain-of-Thought (CoT) reasoning with self-debugging results in the highest token usage but only

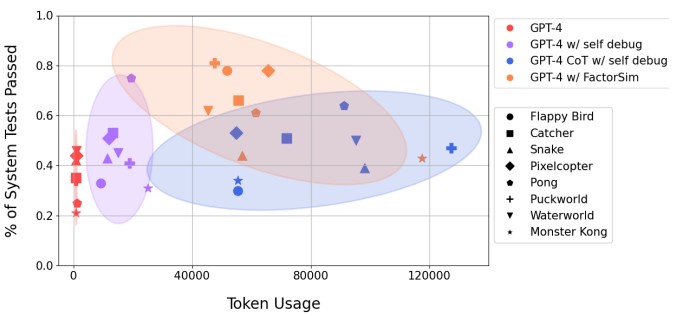

Figure 3: Performance and token usage analysis of GPT-4-based methods. Ellipses correspond to 90% confidence intervals for each algorithm, aggregated over all RL games.

marginally improves accuracy over iterative self-debugging. FACTORSIM achieves the highest accuracy with modest token usage, indicating that the decomposition of tasks reduces the need for extensive iterative debugging.

Empirically, we find that certain prompts, when tested on baselines without the decomposition mechanism in FACTORSIM, are prone to syntax or runtime errors that the LLMs cannot self-debug. This is particularly evident with Llama-3 (vanilla) and Llama-3 self-debug, which perform poorly as they generate highly similar incorrect implementations, ignoring the logic specified in the prompts even when the temperature is set to 1. We hypothesize that this behavior is due to the model having a strong prior for how certain games, like Pong and Flappy Bird, should be implemented, ignoring the prompt specifications. This "mode collapse" issue of LLMs could be caused by over-fitting in the instruction tuning stage [10].

While AgentCoder iteratively refines code, it performs poorly because it relies on a test designer agent and a test executor agent to write quality test cases. However, due to the complexity of the tasks in our benchmark, the test designer agent tends to write incorrect or infeasible tests, leading to negative feedback. This points to FactorSim being an improvement over the standard "role-based" Chain of Thought decompositions, and that it is non-trivial to generate simulations from complex textual specifications.

**Zero-shot Transfer Results**    Additionally, we test FACTORSIM by training a PPO [27] agent on 10 generated environments for 10 million steps and zero-shot test it on the "ground-truth" environment implemented in the original RL benchmark (Figure 4). The rewards are linearly scaled such that 0 corresponds to the performance of a random policy and 1 corresponds to the performance of a PPO agent trained for 10 million steps on the "ground-truth" environment. FACTORSIM achieves notably better zero-shot transfer results as a result of generating code that adheres more closely to the prompt specification. We also observe that the errors FACTORSIM made tend to be more spread out across different components of the simulation. In contrast, many baselines suffer from failure modes concentrated in a specific aspect of the generation (e.g., incorrectly implementing the collision logic) that significantly hampers the ability of a set of generations to facilitate zero-shot transfer.

**Human Study Evaluation**    Automated systems tests cannot holistically capture some aspects of game playability such as rendering a usable user interface. To address this limitation we conducted a human study where users were asked to play the generated games and evaluate their playability. Over 320 human evaluations (40 per game) we find FACTORSIM generates more functional and playable games, compared to the strongest baseline GPT-4 CoT with iterative self-debugging (Figure 5). More details can be found in the Appendix.

## 4.2   Robotics Task Generation

We evaluate on GenSim's [35] 50-task benchmark of robotics tasks in the CLIPort framework [29]. Refer to Figure 6 for an overview of our experimental setting. We compare FACTORSIM with the best-performing methods in generating code that specifies tasks (object states and reward structure) that can be used to train robots. Analogous to the game generation experiment, we use FACTORSIM

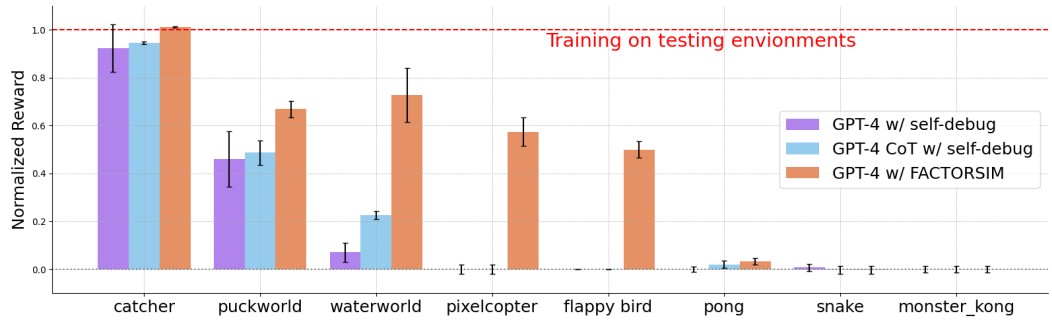

Figure 4: Zero-shot transfer results on previously unseen environments (i.e., environments in the original RL benchmark [33]).

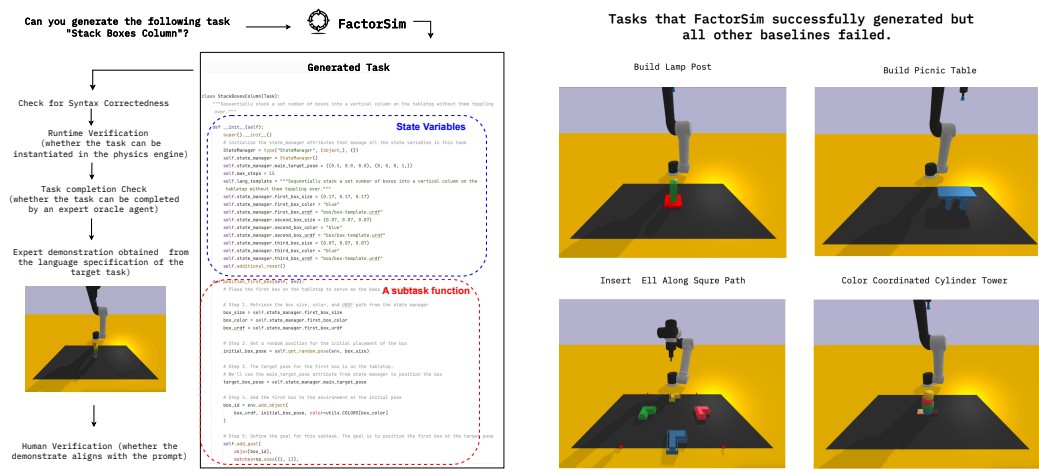

Figure 6: **Left**: an overview of our robotics task generation experimental setting. **Right**: Tasks successfully generated using FactorSim, which all other baselines fail on.

to modularize the code generation process into subtasks and have LLMs generate each subtask using only a set of necessary states as context. More details can be found in the Appendix.

**Baselines & Metrics** We compare our method with the multiple GenSim baselines: vanilla (one-shot), Chain-of-Thought (topdown), and Chain-of-Thought (bottom-up). Adopting the same set of metrics, we evaluate all methods on a sequence of pass rates on "syntax correctness", "runtime-verified", and "task completed". A generated task is considered "completed" if a coded oracle agent could collect 99% of the total reward half of the time.

We empirically found that the "task completion rate" is an imperfect metric for evaluating the success of a generated task. A task deemed "complete" by the oracle agent may fail to adhere to the prompt. For example, when asked to generate a task "build a wheel," a method might

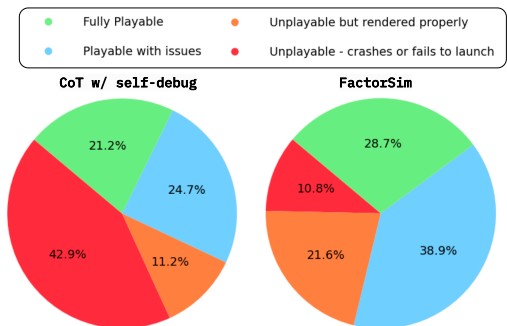

Figure 5: Human evaluation results on the generated simulations of FACTORSIM and the strongest baseline (i.e., GPT-4 CoT w/ self-debug), aggregated over all 8 RL games.

produce a task specification that involves rearranging blocks into a structure that does not resemble a wheel. To address this, we introduced a metric of the "human pass rate". This involved manually inspecting runtime-verified tasks to determine if the task specifications aligned with the prompt descriptions (see Appendix).

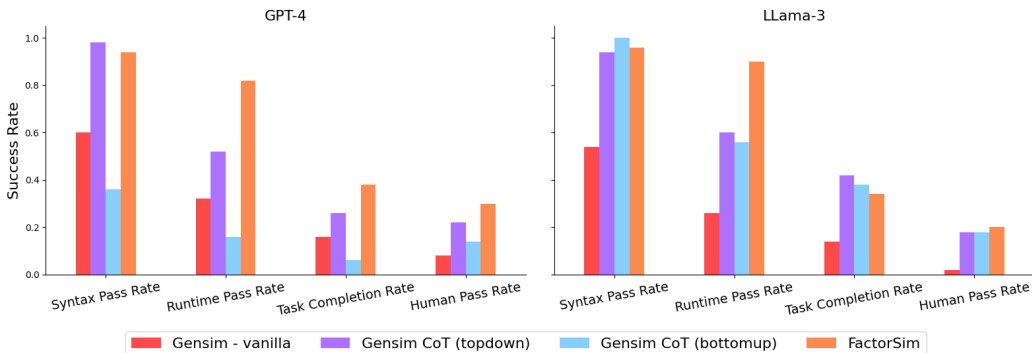

Figure 7: Performance of FACTORSIM and GenSim [35] baselines in generating robotic tasks.

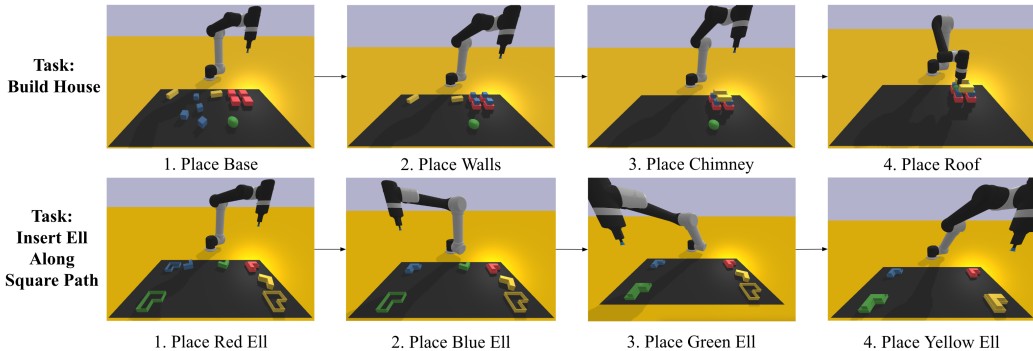

Figure 8: This figure illustrates two input task prompts and the corresponding sequence of subtasks decomposed by FACTORSIM.

**Results & Discussion**  FACTORSIM outperforms baselines in generating tasks with a higher runtime pass rate and better human evaluation scores, indicating improved prompt alignment (Figure 7). Task completion rates are generally low for all methods due to the limitation of the oracle agent. For example, tasks like "Build Ball Pit" (fill a container with balls to create a ball pit) often fail because the balls roll out of the visible area of the oracle agent, not because the generated task is invalid. FACTORSIM performs particularly well on tasks that specify spatial relationships (e.g., "on top of," "left of," "outside of") between objects, such as the "build House" example in Figure 8. This improvement is likely due to the decomposition process, where for each step, instead of addressing a combination of multiple spatial relations all at once, FACTORSIM attends to a smaller context, allowing each spatial relation to be addressed separately.

## 5 Conclusion & Future Work

We have proposed FACTORSIM as an approach to generate full simulations as code that can train agents while adhering to detailed design requirements specified as a text prompt. We also introduce a benchmark suite of eight RL environments to evaluate generative simulation methods.

Generating complex simulations in code is challenging, and we anticipate numerous opportunities to extend the simulation generation process. There is substantial room to address larger-scale, more complex games, and robotics environments that require code bases beyond what can be used effectively in the context window of existing LLMs. We also see great potential to accelerate RL agent training by generating code that can be accelerated on GPU devices. Our robotic simulation results will benefit from further investigations to demonstrate transfer to real-world environments. We have only addressed single-agent simulations, leaving the extension of our method to multi-agent settings to future work. In the future, we also plan to incorporate information from the agent training process to automatically modify the generated simulation environment for enhanced agent learning and generalization. Taken together, we believe the generation of full simulations as code will be an important step toward enhancing the capabilities of LLMs to support the development of generalized RL agent policies.

## Acknowledgments and Disclosure of Funding

This work was in part supported by the Stanford Institute for Human-Centered Artificial Intelligence (HAI), the Stanford Center for Integrated Facility Engineering (CIFE), NSF CCRI #2120095, AFOSR YIP FA9550-23-1-0127, ONR N00014-23-1-2355, ONR YIP N00014-24-1-2117, and Google.

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

## A  Societal Impact

This work can be applied broadly to many types of simulations, including robotics, autonomous vehicles, and other autonomous systems. Such systems have the potential for both positive and negative societal impact (e.g., harmful dual use). As researchers, we must critically evaluate such applications and promote beneficial ones. In this work we have focused on simulations with potential positive social impact, particularly in supporting the development of robots able to operate in human environments like households or manufacturing facilities.

The methods we present generate simulations that can be used to train agents to perform tasks. One risk with generated simulations is for training agents in an unintended manner. By generating simulations specified as code we mitigate this concern by making the behavior of the simulation explicit and inspectable by humans. Further, our approach is better able to guide LLMs to generate code that matches input design specifications compared to baseline methods, reducing the risk of LLMs inadvertently producing undesirable functionality. We believe this can help enhance the reliability of generated simulations while offering strong editing and control capabilities to humans.

The potential negative environmental impact of the compute for using our technique is small. We have shown our technique consumes less tokens than comparable methods to yield equally good results. Thus our method can be seen as a way to reduce computational needs when using LLMs for tasks like creating simulations. Compared to systems that use a neural world model our approach benefits from the relatively lower computational costs of running simulations in code compared to running large neural models for simulation.

## B  Additional details of our experiments

All experiments are done on a workstation with 8 Nvidia A40 GPUs and 1008G of RAM. For our code generation experiments, one generation (i.e., generation of one training environment) takes around 30 seconds to 5 minutes. For our Reinforcement Learning experiments, one trial of training (i.e. training on a set of environments for 10M steps in total) takes around 3-5 hours to complete. In all of our experiments, GPT-4 refers to the OpenAI's "gpt-4-1106-preview" model, GPT-3.5 refers to OpenAI's "gpt-3.5-turbo" model, and Llama-3 refers to the open-sourced "meta-llama-3-70b-instruct" model that can be found on huggingface.

For the zero-shot transfer RL experiment, we supply all methods with the reference "controller" (i.e., the same key press/mouse click leads to the same thing). We do this because language descriptions of such can be very ambiguous (e.g., a description "a key press leads the bird to flap its wings" can imply a change in position, velocity, or acceleration). In our experiments, we generate 10 environments and filter out those that cannot be properly executed with a random policy.

All the RL experiments are implemented in RLLib [20] [8]. The PPO agent is trained with a batch size of 10,000, and an SGD minibatch size of 2048. Our agent used a fully connected network with hidden layers of sizes (4, 4) and post-FCNet hidden layers of size 16, all employing ReLU activation functions. The policy network uses an LSTM with a cell size of 64 to incorporate previous actions but not previous rewards. Over the course of the 10 million training steps, 20 checkpoints were saved, with the best zero-shot performance on the testing environment reported.

## C  Additional details of FACTORSIM

We provide code in the supplementary material. Here we provide the prompts used in FACTORSIM.

Listing 1: The first decompositional prompt used in FACTORSIM.

```
Given an unstructured game description, decompose the game's specification into a set of
    steps or modules. Each step or module should contain at most one input event
    handling, one state transitional logic, and one rendering logic.
If we model the game as a Markov Decision Process (MDP), the steps, or modules of the
    game, should share as little state variables as possible.

Please provide the response in the following format:
```json
```

---

[8]https://docs.ray.io/en/latest/rllib/index.html

```
{{
    "steps": [
        "Step 1: Describe the first module/step",
        "Step 2: Describe the second module/step",
        "Step 3: Describe the third module/step",
        ...
    ],
    "explanations": "To make sure the decomposition don't miss any important game
        mechanics, explain where each module/step fits in the game's logic."
}}
```

For example:
```json
{{
    "steps": [
        "Step 1: Introduce a balloon asset rendered as a blue rectangle. Implement
            gravity so that the balloon is by default always falling down. Allow users
            to move the balloon with arrow keys.",
        "Step 2: Implement a scoring system that rewards the human player for returning
            the ball back (i.e. making contact with the ball). Display the current score
            at the top of the screen. Display a 'Game Over!' message when one player
            reaches a score of 5 and provide an option for the player to restart the
            game after the 'Game Over' screen is displayed.",
        ...
        "Step 5: Implement a 'Game Over' screen that is displayed when the player reaches
            a score of 5. Allow the player to restart the game from this screen."
    ],
    "explanations": "The balloon asset is the main character of the game. Its rendering
        and movement are handled in the first module. The scoring system is implemented
        in the second module to keep track of the player's progress. The 'Game Over'
        logic is implemented in the last module to provide a clear end to the game."
}}

The unstructured specification of the game is:
\{game_specification\}

Please provide the structured steps in the format of the JSON above.
- Ensure that each step is a separable part of the game that can be implemented as
    independently as possible.
- You most likely don't need to decompose the game into more than 5 steps. However, the
    most important thing is to ensure that all the steps accurately describe the game's
    implementation.
- The most important thing is to make sure that the decomposition does not miss any logic
    step (e.g., the balloon should not be able to go off the screen).
- Note that the order of the steps is the order that these modules will be called in the
    game loop. Ensure that the game described can be implemented sequentially. For
    example, the reset position logic should be implemented after the collision
    detection logic.
```

Listing 2: The state context selection prompt used in FACTORSIM.

```
The game designer is building a single-player game in Pygame by modeling this game as a
    Markov Decision Process (MDP).
Your task is to identify and compile a list of relevant state variables to implement a
    specific feature requested by the game designer.
The game already has the following state space implementation:
```python
import pygame
import sys
import random

{state_manager_code}

    # new variables will be added here:
    # variable_description
    self.{{variable_name}} = {{variable_type}}({{value}})
```

Please provide the state variables in the following format within a JSON object:
```json
{{
    "relevant_state_variables": [
        {{
            "variable_name": "Name of the variable",
        }},
        ...
    ],
```

```
        "new_state_variables": [
            {{
                "variable_description": "Description of the variable",
                "variable_name": "Name of the variable",
                "variable_type": "Type of the variable, one of {{int, float, str, bool, tuple
                    , list}}",
                "variable_value": "Value of the variable, e.g. 100, 0.5, 'balloon', tuple
                    ((255, 0, 255)), True, [10, 50], [{{'x': 100, 'y': 200}}]",
            }},
            ...
        ]
}}
```

```

The game designer's request is: {query}.

Here are the dos and don'ts for this request:
- The list "relevant_state_variables" should contain the names of the existing state
    variables that are relevant to the game designer's request.
- Please return a single list of state variables that contains both existing variables
    that you think are relevant and new state variables.
- A software engineer will later implement this request by implementing a function that
    takes these variables as input, so ensure all the variables needed to implement the
    request are included.
- It is okay to include variables that don't end up being used in the implementation
    because redundant state variables will be filtered out later.
- Please provide all rendering variables (e.g., size, color) if there are components to
    be rendered. Color should never be white since the background is white.
- Don't provide Sprite, Surface, or Rect variables. We will handle these variables later.
- Don't introduce variables using existing variables (e.g., self.bird_size = self.
    pipe_size/2), all state variables should be independent of each other.
- Always provide a default value even if a state variable should be chosen randomly. The
    randomness will be implemented later.
- "variable_value" should never to empty like []. Always provide a non-empty default
    value so the software engineer can infer how the variable can be accessed.
- Do not hallucinate external image files (e.g., .png, .jpg) or sound effects(e.g., mp3).
- Prioritize reusing the existing state variables as much as possible. For example, if we
    have "position_x" and "position_y" of a character, do not give me another variable
    "positions" in a list format.
- Additionally, you may add new state variables to the list "new_state_variables" if
    necessary. Please only create new state variables if necessary.
"""
```

Listing 3: The prompt for the **Controller** component (as defined in the Model-View-Controller) utilized in the FACTORSIM.

```
The game designer is building a single-player game in Pygame by modeling this game as a
    Markov Decision Process (MDP). Your task is to detect key/mouse input and update the
    state variables accordingly according to a feature requested by the game designer.
The game has the following implementation already:
```python
import pygame
import sys
import random

{state_manager_code}

# existing input event handling functions
{existing_implementation}
# the new logic function will be here
# if the function is already implemented, it will be replaced with the new implementation

def main():
    state_manager = StateManager()
    running = True
    while running:
        event = pygame.event.poll()
        if event.type == pygame.QUIT:
            running = False
        # {{function_description}}
        {{function_name}}(state_manager, event)
    pygame.quit()

if __name__ == "__main__":
    pygame.init()
    main()
```
```

```
To ensure the implementation is correct, please also implmeent an unit test for the
    function. Please implement the following request from the game designer and return
    your answer in the following format:
```json
{{
    "function_name": "{function_name}",
    "function_description": "{function_description}",
    "function_implementation": "the pygame implementation of the function, including the
        first line of the function definition",
    "unit_test": "the unit test code for the function"
}}
```

Here are the dos and don'ts for this request:
- Note that the implementation of the function shuold only have two arguments (i.e.
    state_manager and event).
- The function implementation should involve checking user input with event (i.e. event.
    type and event.key).
- Minimize the number of functions added while meeting the game designer's requirements.
    However, make sure to always give the full implementation of the function.
- Include only the essential details requested by the game designer. Do not add things
    that are not requested.
- Please use the state variables defined in the state manager. Do not introduce new state
    variables.
- Only KEYDOWN events will be detected. Do not rely on KEYUP events.
- Check for index out of bounds errors with any lists or arrays being used.
- Check for divide-by-zero errors.
- Do not leave any code incomplete. Do not leave placeholder values. Do not provide
    demonstration code implementation. Be sure all code is fully implemented.
```

Listing 4: The prompt for the **Model** component (as defined in the Model-View-Controller) utilized in the FACTORSIM.

```
The game designer is building a single-player game in Pygame by modeling this game as a
    Markov Decision Process (MDP). Your task is to define and code new state transition
    functions according to the feature requested by the game designer.
The game has the following implementation already:
```python
import pygame
import sys
import random

{state_manager_code}

# existing state transition functions
{existing_implementation}
# the new function will be here
# if the function is already implemented, it will be replaced with the new implementation

def main():
    state_manager = StateManager()
    running = True
    while running:
        event = pygame.event.poll()
        if event.type == pygame.QUIT:
            running = False
        # {{function_description}}
        {{function_name}}(state_manager)
    pygame.quit()

if __name__ == "__main__":
    pygame.init()
    main()
```

To ensure the implementation is correct, please also implmeent an unit test for the
    function. Please implement the following request from the game designer and return
    your answer in the following format:
```json
{{
    "function_name": "{function_name}",
    "function_description": "{function_description}",
    "function_implementation": "the pygame implementation of the function, including the
        first line of the function definition",
    "unit_test": "the unit test code for the function"
}}
```
```

```
Here are the dos and don'ts for this request:
- Only implement things that pertain to updating the state variables. Other aspects of
    the game like input event handling and UI components will be handled separately.
- Include only the essential details requested by the game designer. Do not add things
    that are not requested.
- These state transition functions will be called in every iteration of the main game
    loop. If you want to add a conditional logic to the function, please implement it in
     the function itself.
- Note that this new function will be added to the end of the list of state transition
    functions.
- Please use the state variables defined in the state manager. Do not introduce new state
     variables.
- Check for index out of bounds errors with any lists or arrays being used.
- Check for divide-by-zero errors.
- Do not leave any code incomplete. Do not leave placeholder values. Do not provide
    demonstration code implementation. Be sure all code is fully implemented.
```

Listing 5: The prompt for the **View** component (as defined in the Model-View-Controller) utilized in the FACTORSIM.

```
The game designer is building a single-player game in Pygame by modeling this game as a
    Markov Decision Process (MDP). Your task is to add rendering functions that decide
    how state variables are rendered as UI components on the screen, according to the
    feature requested by the game designer.
The game has the following implementation already:
'''python
import pygame
import sys
import random

{state_manager_code}

# existing rendering functions
{render_code}
# the new function will be here
# if the function is already implemented, it will be replaced with the new implementation

def main():
    state_manager = StateManager()
    clock = pygame.time.Clock()
    running = True
    while running:
        action = pygame.event.poll()
        if action.type == pygame.QUIT:
            running = False

        # all the code for state transitional logics
        # omitted for brevity

        # Fill the screen with white
        state_manager.screen.fill((255, 255, 255))
        # all the code for rendering states as UI components
        # {{function_description}}
        {{function_name}}(state_manager)
        pygame.display.flip()
        state_manager.clock.tick(state_manager.fps)

    pygame.quit()

if __name__ == "__main__":
    pygame.init()
    pygame.display.set_caption("")
    main()
'''

To ensure the implementation is correct, please also implmeent an unit test for the
    function. Please implement the following request from the game designer and return
    your answer in the following format:
'''json
{{
    "function_name": "{function_name}",
    "function_description": "{function_description}",
    "function_implementation": "the pygame implementation of the function, including the
        first line of the function definition",
    "unit_test": "the unit test code for the function"
}}
'''
```

| Target Tasks | FactorSim | | | | GenSim - vanilla | | | | GenSim Chain of Thought (topdown) | | | | GenSim Chain of Thought (bottomup) | | | |
|---|---|---|---|---|---|---|---|---|---|---|---|---|---|---|---|---|
| | Syntax | Runtime | Task | Human | Syntax | Runtime | Task | Human | Syntax | Runtime | Task | Human | Syntax | Runtime | Task | Human |
| BuildDogHouse | yes | no | no | no | no | no | no | no | yes | yes | no | no | yes | yes | no | no |
| BuildLampPost | yes | yes | yes | yes | yes | no | no | no | yes | no | no | no | yes | no | no | no |
| BuildNewsstand | yes | yes | no | no | yes | no | no | no | yes | no | no | no | yes | no | no | no |
| BuildBench | yes | no | no | no | yes | no | no | no | yes | no | no | no | yes | no | no | no |
| BuildPicnicTable | yes | yes | yes | yes | no | no | no | no | yes | yes | no | no | yes | yes | no | no |
| BuildBicycleRack | yes | no | no | no | yes | no | no | no | yes | no | no | no | yes | no | no | no |
| BuildPicnicBasket | yes | no | no | no | yes | no | no | no | yes | yes | no | no | yes | yes | no | no |
| BuildCylinderStructure | yes | yes | yes | yes | no | no | no | no | yes | no | no | no | yes | no | no | no |
| BuildBridge | yes | yes | yes | no | yes | no | no | no | yes | no | no | no | yes | no | no | no |
| BuildCar | yes | yes | no | no | yes | yes | no | no | yes | yes | yes | no | yes | yes | yes | no |
| BuildTwoCircles | yes | yes | no | no | yes | no | no | no | no | no | no | no | no | no | no | no |
| BuildWheel | no | no | no | no | yes | no | no | no | yes | yes | no | no | yes | yes | no | no |
| Aggregated Pass Rate | **92%** | **58%** | **33%** | **25%** | 75% | 8% | 0% | 0% | 92% | 42% | 8% | 0% | 92% | 42% | 8% | 0% |

Table 2: **Additional Experimental Results**: FactorSim outperforms other Chain of Thought baselines by a large margin on assembly tasks. *Syntax* indicates the task passes the syntax check. *Runtime* indicates that the task can run in the physics simulator. *Task* indicates whether the task can be completed by the oracle agent. *Human* indicates whether the completed task matches the input prompt specification.

```
Here are the dos and don'ts for this request:
- Only implement things that pertain to how state variables are rendered as UI components
    on the screen. Other aspects like input event handling and state transition will be
    handled separately.
- Please make sure that all of the state variables remain unchanged in the rendering
    functions.
- Include only the essential details requested by the game designer. Do not add things
    that are not requested.
- These rendering functions will be called in every iteration of the main game loop. If
    you want to add a conditional logic to the function, please implement it in the
    function itself.
- Note that the background color of the screen is white so white UI components will not
    be visible. Do not fill the screen with white again in the rendering functions.
- Note that the new function will be added to the end of the list of rendering functions.
- Please use the state variables defined in the state manager. Do not introduce new state
    variables.
- Check for index out of bounds errors with any lists or arrays being used.
- Check for divide-by-zero errors.
- Do not call pygame.display.set_mode in UI functions. Only call it once outside of any
    UI function that is called multiple times.
- Do not leave any code incomplete. Do not leave placeholder values. Do not provide
    demonstration code implementation. Be sure all code is fully implemented.
```

# D  Additional details and results for the robotics task generation experiment

In this section, we provide the prompts we used for FACTORSIM in the robotics task generation experiment. The prompts for the baselines can be found in the GenSim paper[9] [35].

To conduct human evaluation, we begin by observing the oracle agent attempting to solve the task. If the oracle agent successfully completes the task, we then assess whether the resulting goal states align with the input task prompt. If the oracle agent fails to solve the task, we investigate the reason for the failure. Often, the cause is apparent, such as the target container being too small or not having the right color of objects for the task. These are marked as failures. For cases where it is clear that the limitation lies in the oracle agent's ability, or when the reason for failure is not immediately apparent, we manually inspect the code for the task specification and base our decision on both the code and our observation of the oracle agent's attempt at solving the task.

Listing 6: The chain of thought prompt.

```
The game designer is building a single-player game in Pygame by modeling this game as a
    Markov Decision Process (MDP). Your task is to detect key/mouse input and update the
    state variables accordingly according to a feature requested by the game designer.
The game has the following implementation already:
```python
import pygame
import sys
import random

{state_manager_code}
```

---

[9]https://github.com/liruiw/GenSim

```
# existing input event handling functions
{existing_implementation}
# the new logic function will be here
# if the function is already implemented, it will be replaced with the new implementation

def main():
    state_manager = StateManager()
    running = True
    while running:
        event = pygame.event.poll()
        if event.type == pygame.QUIT:
            running = False
        # {{function_description}}
        {{function_name}}(state_manager, event)
    pygame.quit()

if __name__ == "__main__":
    pygame.init()
    main()
```

To ensure the implementation is correct, please also implmeent an unit test for the
    function. Please implement the following request from the game designer and return
    your answer in the following format:
```json
{{
    "function_name": "{function_name}",
    "function_description": "{function_description}",
    "function_implementation": "the pygame implementation of the function, including the
        first line of the function definition",
    "unit_test": "the unit test code for the function"
}}
```

Here are the dos and don'ts for this request:
- Note that the implementation of the function shuold only have two arguments (i.e.
    state_manager and event).
- The function implementation should involve checking user input with event (i.e. event.
    type and event.key).
- Minimize the number of functions added while meeting the game designer's requirements.
    However, make sure to always give the full implementation of the function.
- Include only the essential details requested by the game designer. Do not add things
    that are not requested.
- Please use the state variables defined in the state manager. Do not introduce new state
    variables.
- Only KEYDOWN events will be detected. Do not rely on KEYUP events.
- Check for index out of bounds errors with any lists or arrays being used.
- Check for divide-by-zero errors.
- Do not leave any code incomplete. Do not leave placeholder values. Do not provide
    demonstration code implementation. Be sure all code is fully implemented.
```

Listing 7: The state change prompt.

```
**Objective:** Create a Python program that generates state variables for a robot
    simulation task. The state variables should be able to depict the final target
    environment for this task.

**Task Details:**
- Task: `TARGET_TASK_NAME`
- Goal: `TARGET_TASK_DESCRIPTION`
- URDFs: `TASK_ASSET_PROMPT`

**Requirements:**
- Return a python function called `get_state_variables` which returns a list of `
    StateVariableData` objects. We should be able to take the result of this function
    and plug them into a robotics environment, and it should display what the final
    target environment will look like for the given task.
- The `StateVariableData` dataclass is defined as follows:
```
@dataclass
class StateVariableData:
    variable_description: str
    variable_name: str
    variable_urdf: str
    variable_size: Tuple[float, float, float]
    variable_color: str
    variable_target_pose: List[Tuple[Tuple[float, float, float], Tuple[float, float,
        float]]]
    variable_amount: str
    static: bool
```

```
‘‘‘

**Important Notes**
- Item sizes and positions must reflect a realistic setting. Ensure logical spatial
    relationships; items should neither overlap nor float unnaturally. Account for the
    items' dimensions and sizes when determining their placement.
- "variable_name" is a unique name for every state variable.
- "variable_size" should be a list of 3 floats, representing the size of the object along
    the three dimension [x, y, z].
- "variable_target_pose" should be a list of poses, where each pose is a 2-element list
    containining the 3-element position vector, and 4-element quaternion rotation. The
    position vectors should be within the following boundaries: [0.25, 0.75] for the x-
    axis, [-0.5, 0.5] for the y-axis, and [0.01, 0.3] for the z-axis. For a single
    object, the target pose should only contain one element, like [[[0.52, 0.02, 0.001],
    [0, 0, 0, 1]]]. For multiple objects, the target pose can contain multiple items,
    like [[[0.52, 0.02, 0.001], [0, 0, 0, 1]], [[0.48, 0.02, 0.001], [0, 0, 0, 1]]].
- "static" describes if an object is something part of the environment and is not
    something the robotic agent should be moving. For example, if the task is to put a
    ball into the container, the container should be static.

**Example Input and Expected Output: **
- Task: 'build_a_car'
- Goal: "Construct a simple car structure using blocks and cylinders."
- URDFs: ["box/box-template.urdf", "cylinder/cylinder-template.urdf"]
- Expected output
‘‘‘python
def get_state_variables() -> List[StateVariableData]:
    car_pose = ((0.5, 0.0, 0.0), (0,0,0,1))  # fixed pose
    base_length = 0.04

    base_target_pose = [(utils.apply(car_pose, (base_length / 2,  base_length / 2, 0.001)
        )), car_pose[1]),
                    (utils.apply(car_pose, (-base_length / 2,  base_length / 2, 0.001)),
                        car_pose[1])]

    wheel_length = 0.12
    wheel_target_poses = [(utils.apply(car_pose, ( wheel_length / 2,  wheel_length / 2,
        0.001)), car_pose[1]),
                            (utils.apply(car_pose, (-wheel_length / 2,  wheel_length / 2,
                                0.001)), car_pose[1]),
                            (utils.apply(car_pose, ( wheel_length / 2, -wheel_length / 2,
                                0.001)), car_pose[1]),
                            (utils.apply(car_pose, (-wheel_length / 2, -wheel_length / 2,
                                0.001)), car_pose[1])]

    return [
        StateVariableData(
            variable_description="blocks used to build the base",
            variable_name="base",
            variable_size=[0.02, 0.04, 0.02],
            variable_urdf="box/box-template.urdf",
            variable_color="red",
            variable_target_pose=base_target_pose,
            variable_amount=2,
            stationary=False,
        ),
        StateVariableData(
            variable_description="wheel to put on the base",
            variable_name="wheel",
            variable_size=[0.02, 0.02, 0.02],
            variable_urdf="cylinder/cylinder-template.urdf",
            variable_color="black",
            variable_target_pose=wheel_target_poses,
            variable_amount=4
            stationary=False,
        ),
        StateVariableData(
            variable_description="body of the car",
            variable_name="body",
            variable_size=[0.04, 0.02, 0.02],
            variable_urdf="box/box-template.urdf",
            variable_color="blue",
            variable_target_pose":[car_pose],
            variable_amount=1
            stationary=False,
        )
    ]
‘‘‘

**Example Input and Expected Output: **
- Task: 'build_a_circle'
```

```python
- Goal: "Construct a circle using 6 red blocks"
- URDFs: ["block/block.urdf"]
- Expected output
'''python
def get_state_variables() -> List[StateVariableData]:
    block_size = (0.04, 0.04, 0.04)

    red_circle_poses = []
    circle_radius = 0.1
    circle_center = (0, 0, block_size[2] / 2)
    angles = np.linspace(0, 2 * np.pi, 6, endpoint=False)
    circle_pose = ((0.4, 0.3, 0.0), (0, 0, 0, 1))  # fixed pose

    # Define initial and target poses for the red and blue circles.
    for angle in angles:
        pos =  (circle_center[0] + circle_radius * np.cos(angle),
                circle_center[1] + circle_radius * np.sin(angle),
                circle_center[2])
        block_pose = (utils.apply(circle_pose, pos), circle_pose[1])
        red_circle_poses.append(block_pose)

    return [
        StateVariableData(
            variable_description="blocks used to build the base",
            variable_name="base",
            variable_size=block_size,
            variable_urdf="block/block.urdf",
            variable_color="red",
            variable_target_pose=red_circle_poses,
            variable_amount=10,
            stationary=False,
        ),
    ]
'''
```

Listing 8: The subtask code generation prompt .

```
**Objective:** You are designing a training plan for a robotic arm in a simulation
    environment to complete a task 'TARGET_TASK_NAME'. This task will be completed in
    multiple subtasks from the subtask list. Each subtask manages the robotic arm to
    move the composition of assets from one state to another, ultimately achieving the
    ideal state that completes the task.
For this step, you are asked to generate the subtask function for 'SUBTASK_NAME'. It
    involves 'SUBTASK_DESCRIPTION'. Refer to the existing template and use the existing
    variables to inform your subtask creation. You will generate Python code for {
    state_variable_for_SUBTASK_NAME} and {subtask_code_for_SUBTASK_NAME}, and return the
     code with the given JSON format.

**Task Overview:**
- Task Name: 'TARGET_TASK_NAME'
- Task Description: 'TARGET_TASK_DESCRIPTION'
- All Subtask Descriptions: 'TARGET_SUBTASK_LIST'
- Current Subtask Name: 'SUBTASK_NAME'
- Current Subtask Description: 'SUBTASK_DESCRIPTION'

**Existing template**
SUBTASK_CODE_TEMPLATE

**Subtask Requirements:**
- Generate Python code addressing the specified subtask based on the provided description
     and initial variables.
- Adhere to guidelines: use specified APIs you just reviewed, avoid unknown libraries,
    and comment on the code for clarity.
- Ensure all state manager variables used in the subtask code are already defined.

**Subtask function important Notes:**
- Only one 'self.add_goal' should be used, and it should not be used on state variables
    marked with static=True.
- The 'matches' argument in the method called 'self.add_goal' should always be a numpy
    array.
- Do not use libraries, functions, and assets that you don't know.
- Do not create additional functions inside the subtask function, only return one
    function.
- Do not add state variables marked as "static = True" to the environment using 'env.
    add_object'.
```

- When passing in color using `env.add_object`, remember to pass it using utils.COLORS[
    item_color]. Good example is: `env.add_object(base_block_urdf, base_block_pose,
    color=utils.COLORS['red'])`. Bad example is: `env.add_object(base_block_urdf,
    base_block_pose, 'red')`
- In the subtask code, you need to create both the initial pose and the target pose. The
    Initial pose should NOT be very close to or be the same as the target pose. Initial
    pose determines where the object will be placed before the training starts. Target
    pose determines the ideal pose where the robot arm will receive reward if placed
    right in the simulation. You should prioritize using the target_pose from the state
    manager variables.
- Only use get_random_pose for initial pose(position and rotation). `get_random_pose(env,
    obj_size)` gets random collision-free object pose within workspace bounds. param
    obj_size: (3, ) contains the object size in x,y,z dimensions. return: translation
    (3, ), rotation (4, ). You should pass the obj_size directly to the get_random_pose
    function, instead of its single element. Good example is: self.get_random_pose(env,
    block_size). Bad example is: self.get_random_pose(env, block_size[0]) or env.
    get_random_pose(block_size).
- If you are generating a pose from `get_random_pose`. It will be within bounds. You don'
    t need to check it using other helper function like `is_pose_within_bounds`. Do not
    use `np.copy` to copy it.
- If you need initial rotation, use `get_random_pose` to get the `pose` first. Then use `
    pose[1]` as rotation. Do not initiate it in other ways, a bad example is `rotation =
    np.float32(p.getMatrixFromQuaternion(pose[1])).reshape(3, 3)`.
- DO NOT USE `pose = p.getBasePositionAndOrientation(object_id)`. It's for environment
    simulation.
- DO NOT USE other unlisted way to create `pose`.
- In self.add_goal, make sure to set "step_max_reward=1./SUBTASK_COUNT"
- Each `env.add_object` call will create a new object id. If only one object is needed in
    this subtask, then pass in `objs = [object_id]` in `self.add_goal`. If multiple
    objects are needed, create a list that contains all the needed object ids then pass
    in the list: `objs = object_id_list`.
- Make sure you include all the arguments to `self.add_goal`: `objs`, `matches`, `
    targ_poses`, `replace`, `rotations`, `metric`, `params`, `step_max_reward`, `
    language_goal`.
- For `self.add_goal`'s argument `matches`, it should be `matches=np.ones((n, n))`. `n`
    represents the total amount of objects added to the `env`. E.g. `env.add_object` was
    called 4 times then it should be set as `matches=np.ones((4, 4))`.
- You have been given all the task variables for creating the subtask. Do not assume any
    unknown variables.
- Only three functions are available from `utils`: `utils.apply`, `utils.
    quatXYZW_to_eulerXYZ` and `utils.COLORS`. Do not make up any other functions from `
    utils`.
- Do not include triple quotes (""") in your code, only use `#` for comments.
- If the asset of this subtask involves `zone`, make sure that pose of the zone should
    not be moved, it's usually used for creating target position for other items. E.g:
`zone_size = [0.12, 0.12, 0]
zone_urdf = 'zone/zone.urdf'
zone_colors = ['yellow', 'blue', 'green']
zone_poses = []
for color in zone_colors:
    zone_pose = self.get_random_pose(env, zone_size)
    env.add_object(zone_urdf, zone_pose, 'fixed', color=utils.COLORS[color])
    zone_poses.append(zone_pose)`
- If the subtask involves updating the rotation, you may call `utils.quatXYZW_to_eulerXYZ
    `. Here's what this function do:
`def quatXYZW_to_eulerXYZ(quaternion_xyzw):  # pylint: disable=invalid-name
    """Abstraction for converting from quaternion to a 3-parameter rotation.

    This will help us switch which rotation parameterization we use.
    Quaternion should be in xyzw order for pybullet.

    Args:
      quaternion_xyzw: in xyzw order, tuple of 4 floats

    Returns:
      rotation: a 3-parameter rotation, in xyz order, tuple of 3 floats
    """
    q = quaternion_xyzw
    quaternion_wxyz = np.array([q[3], q[0], q[1], q[2]])
    euler_zxy = euler.quat2euler(quaternion_wxyz, axes='szxy')
    euler_xyz = (euler_zxy[1], euler_zxy[2], euler_zxy[0])
    return euler_xyz
`

**Example Input and Expected Output:**
- Task Name: `build_a_car`
- Task Description: `Construct a simple car structure using blocks and cylinders.`
- Subtask List: `["build_car_base: Build the base of the car in the simulation
    environment.", "build_car_body: Build the body of the car in the simulation
    environment.", "build_car_wheels: Build the wheels of the car in the simulation
    environment."]`

```
- Subtask Name: 'build_car_base'
- Subtask Description: 'Build the base of the car in the simulation environment.'
- Existing template:

import numpy as np
from cliport.tasks.task import Task
from cliport.utils import utils

class MyBuildCar(Task):
    """Construct a simple car structure using blocks and cylinders."""

    def __init__(self):
        super().__init__()
        # initialize the state_manager attributes that manage all the state variables in
            this task
        StateManager = type('StateManager', (object,), {})
        self.state_manager = StateManager()
        self.state_manager.main_target_pose = [[0.5, 0.0, 0.0], [0, 0, 0, 1]]
        self.max_steps = 15
        self.state_manager.base_size = (0.04, 0.08, 0.02)
        self.state_manager.base_color = "green"
        self.state_manager.base_urdf = "box/box-template.urdf"
        self.state_manager.base_amount = 2
        self.state_manager.anchor_base_poses = [((0.52, 0.02, 0.001), (0, 0, 0, 1)),
            ((0.48, 0.02, 0.001), (0, 0, 0, 1))]
        self.additional_reset()

{placeholder_subtask_code}

    def reset(self, env):
        super().reset(env)
        self.place_first_step(env)

- Expected output:
```python
def build_car_base(self, env):
    # Build the base of the car in the simulation environment. Let's solve this problem
        step-by-step.

    # Step 1. Retrieve the variables(the car's pose, the base length, and the base size)
        to initialize the car base building.
    base_size = self.state_manager.base_size
    base_color = self.state_manager.base_color
    base_amount = self.state_manager.base_amount
    anchor_base_poses = self.state_manager.anchor_base_poses

    # Step 2. Setting up the base block URDF.
    base_urdf_path = self.state_manager.base_urdf
    base_block_urdf = self.fill_template(base_block_urdf, {'DIM': base_size})

    # Step 3: Adding base blocks to the scene
    base_blocks = []
    for idx in range(2):
        base_block_pose = self.get_random_pose(env, base_size)
        base_block_id = env.add_object(base_block_urdf, base_block_pose, color=utils.
            COLORS['red'])
        base_blocks.append(base_block_id)

    # Step 4: Setting the goal to create the base of the car by positioning two red
        blocks side by side.
    self.add_goal(
        objs=base_blocks,
        matches=np.ones((base_amount, base_amount)),
        targ_poses=anchor_base_poses,
        replace=False,
        rotations=True,
        metric='pose',
        params=None,
        step_max_reward=1./3,
        language_goal="Firstly, create the base of the car by positioning two red blocks
            side by side."
    )
```

REFLECT_PROMPT

**Output Format:**
- Use 'SUBTASK_NAME' as the function name.
- IMPORTANT doublecheck your code is following everything in important notes.
```

# E   Additional details for the proposed generative simulation benchmark

This section provides the prompts for all 8 RL games in the benchmark.

Listing 9: The prompt for the game *Catcher*.

```
Create a catcher character, represented as a rectangle, positioned at the bottom and the
    middle of the screen.
Allow the player to control the catcher's horizontal movement using the left and right
    arrow keys on the keyboard.
There should always be exactly one ball on the screen at all times. The ball should be
    visually distinct and easily recognizable.
Make the ball move downwards at a steady pace towards the catcher. The speed can be
    constant or increase gradually as the game progresses.
Detect collisions between the catcher and the ball. When the catcher catches a ball,
    increment the player's score, spawn a new ball, and display this score in the top-
    left corner of the screen.
Give the player a 3 lives. Each time a ball is missed by the catcher and reaches the
    bottom of the screen, decrease the player's life count by one.
End the game when the player's lives reach zero. Display a "Game Over!" message and
    temporarily halt gameplay but dont terminate the game.
Provide an option for the player to click the screen to restart the game after the "Game
    Over" screen is displayed.
Continuously generate new balls after each catch or miss, ensuring endless gameplay.
    Optionally, increase the game's difficulty gradually by speeding up the ball's fall
    or reducing the size of the catcher as the player's score increases.
```

Listing 10: The prompt for the game *Flappy Bird*.

```
Create a bird character, visually represented as a simple rectangle within the game
    window.
Introduce gravity, causing the bird to continuously fall slowly.
Allow the bird to 'jump' or accelerate upwards in response to a player's mouse click,
    temporarily overcoming gravity.
Periodically spawn pairs of vertical pipes moving from right to left across the screen.
    Each pair should have a gap for the bird to pass through, and their heights should
    vary randomly.
If the bird makes contact with the ground, pipes or goes above the top of the screen the
    game is over.
Implement the following scoring system: for each pipe it passes through it gains a
    positive reward of +1. Each time a terminal state is reached it receives a negative
    reward of -1.
When the game ends, display a "Game Over!" messagea and stop all the motion of the game.
Show the current score in the top-left corner of the screen during gameplay.
Ensure the game has no predefined end and that new pipes continue to generate,
    maintaining consistent difficulty as the game progresses.
```

Listing 11: The prompt for the game *Snake*.

```
Create a snake character represented as a series of connected pixels or blocks. Initially
    , the snake should be a single block (i.e. the head) that moves in a specific
    direction within the game window.
Allow the player to control the snake's movement using arrow keys. The snake should be
    able to turn left or right, but it should not be able to move directly backward. Eg.
     if its moving downwards it cannot move up.
The movement of the snake should be continuous in the current direction until the player
    provides new input. Ensure that the snake moves one grid unit at a time.
Implement a basic food system where one food item appears randomly on the screen.
When the snake consumes the food by moving over or colliding with it, the snake's length
    increases, and the player earns points. It recieves a positive reward, +1, for each
    food the head comes in contact with. While getting -1 for each terminal state it
    reaches.
If the head of the snake comes in contact with any of the walls or its own body, the game
    should end.
Incorporate a scoring system, displaying the current score on the screen during gameplay.
    The score should increase each time the snake consumes food.
Ensure that the game has no predefined end, allowing the snake to continue growing and
    the difficulty to increase over time. New food items should appear after the snake
    consumes one.
Provide an option for the player to restart the game after it ends. Display a "Restart"
    option on the game over screen to allow the player to play again.
```

Listing 12: The prompt for the game *Pixelcopter*.

```
Create a copter character represented as a large white square that remains fixed
    horizontally but can ascend and descend vertically within the game window.
```

```
Introduce gravity mechanics, causing the copter to continuously descend slowly. Enable
    the player to ascend when the player clicks the mouse, allowing it to momentarily
    counteract gravity and rise upwards.
Create obstacles in the shape of a cavern. Construct the cavern using a series of
    vertically aligned rectangular barriers positioned at both the bottom and the top of
    the screen. Ensure the adjacent obstacles are of similar length to maintain a
    consistently smooth "tunnel" effect.
Implement collision detection to detect when the copter collides with obstacles or the
    boundaries of the game window, triggering the end of the game upon collision.
Display a "Game Over!" message prominently when the game ends due to a collision, halting
    all movement within the game and prompting the player to restart.
Create a scoring system that rewards the player based on how far the copter travels
    through the maze without colliding with obstacles.
Show the current score in the top left area of the screen.
Ensure the game has no predefined end and that new obstacles continue to generate,
    maintaining consistent difficulty as the game progresses
Allow the player to start a new game after a collision.
```

Listing 13: The prompt for the game *Pong*.

```
Create a paddle character for the human player, represented as a rectangle, positioned on
    the left side of the screen.
Allow the human player to control the paddle's vertical movement using the up and down
    arrow keys. The paddle has a little velocity added to it to allow smooth movements.
Implement a paddle character for the CPU opponent, also represented as a rectangle,
    positioned on the opposite side of the screen.
Introduce a ball that moves across the screen with a speed. The ball should bounce off
    the paddles and the top and bottom walls of the game window.
If the ball goes off the left or right side of the screen, reset its position to the
    center and its direction.
The CPU to control its paddle's vertical movement to autonomously track the ball.
Detect collisions between the ball and the paddles. When the ball collides with a paddle,
    make it bounce off in the opposite direction.
Implement a scoring system that rewards the human player for returning the ball back (i.e
    . making contact with the bal).
Display the current score at the top of the screen. Ensure the game has no predefined end
    , allowing for continuous play.
Display a "Game Over!" message when one player reaches a score of 5 and provide an option
    for the player to restart the game after the "Game Over" screen is displayed.
```

Listing 14: The prompt for the game *Puckworld*.

```
Create an agent character, visually represented as a blue circle, positioned on the
    screen. The agent should be movable in any direction based on user input.
Implement a green dot that moves randomly around the screen, serving as the target for
    the agent to navigate towards.
Introduce a red puck, a larger entity that slowly follows the agent's movements on the
    screen.
Allow the player to control the agent's movement using arrow keys or another specified
    input method.
Implement a scoring system that positively rewards the agent proportionally to the
    closeness between the agent and the green dot, and penalizes the agent for proximity
    to the red puck.
Display the current score in the top-left corner of the screen during gameplay.
Ensure the game has no predefined end, allowing for endless gameplay.
Upon reaching the green dot, relocate it to a new random position, maintaining the
    challenge for the player.
```

Listing 15: The prompt for the game *Waterworld*.

```
Create a player character visually represented as a blue circle that can move freely
    within the game window using arrow keys.
Introduce a dynamic environment with equal number of green and red circles. Make the
    green and red circles move randomly around the screen.
Implement a scoring system, where the player earns points for each green circle captured
    and deduct one for each red circle .
Display the current score in the top-left corner of the screen during gameplay.
When the player captures a circle, make it respawn in a random location on the screen as
    either red or green.
Ensure the game continues until all green circles have been captured. Once all green
    circles are captured, display a "Game Over!" message and stop all motion in the game
    .
Provide an option for the player to restart the game after it ends, creating a loop for
    continuous gameplay.
```

Listing 16: The prompt for the game *Monster Kong*.

```
Write code for a 2d game viewed from the side where the character can walk and jump. Let
    the character move left or right using the 'a' and 'd' keys. Let the character jump
    using the 'w' key.
Create a level by arranging 5 stationary platforms above the ground. Make sure the
    character's jump can reach the platform height.
Let the character stand on the ground or platforms but fall otherwise. Start the player
    on the ground.
Add a princess character that the character must rescue by reaching her position. Place
    the princess on one of the platforms.
Implement fireballs that fall from random places from the top of the screen. Do not let
    the fireballs move through the platforms. These fireballs serve as obstacles that
    the player must avoid.
Touching a fireball should deduct 25 points from the player's score and cause them to
    lose a life. The game ends if the player loses fifteen lives.
Scatter ten coins randomly around the game window that the player can collect for 5
    points each.
Award the player 50 points for rescuing the princess. Move the princess to a random
    platform when the player rescues her.
Display the current score and remaining lives in the game window.
```

# F    Additional details for the human study experiment

In this section, we first provide details for the experiment and then the instructions we gave to human participants in our human study, along with the user interface. We also provide the detailed results of this evaluation for all games in Figure 9.

Human participants were asked to play and evaluate the generated games given the prompt while excluding factors such as aesthetics or difficulty. They rated the games on a scale of 1 to 4, where 4 indicates a fully playable game, 3 is a playable game with some bugs or flaws that hinder gameplay experience, 2 is an unplayable game (i.e., no interactivity) with correctly rendered UI, and 1 is a game that crashes or fails to launch.

Listing 17: The instructions we give to human participants to our human study.

```
Welcome to your user study! Your task is to evaluate AI-generated games.
Select the game you want to generate and click the button "Generate" to generate games.
You might have to wait for the game to load for 5-10 seconds.

Note that the game is intentionally slowed down, making it easier for you to evaluate
    them!
Thus, when you click or press a key/button, the "character" might react slower than you
    expected.
Please click "Random Generate" to generate a game.

- If the game doesn't load (black screen), select "1 - unplayable."
- Please do not consider the difficulty of the game.
- Please don't take aesthetics into account.
- You want to assess whether the UI elements are rendered accurately for gameplay
    purposes while excluding considerations related to aesthetics, overlapping, or
    duplicated UI components.

Given the prompt as shown, please judge the playability of the game.
- 4: Fully Playable: the game is generally playable from start to finish without
    significant bugs.
- 3: Playable but with some flaws: the game is somewhat playable (interactable), but
    there are issues (inaccurate logic or glitches) that impair the gameplay.
- 2: Not playable: no interactivity but the UI seems to be rendered correctly.
- 1: Unplayable: the game cannot be started, or it crashes immediately upon launch.
```

Figure 9: Human study results on all 8 games.

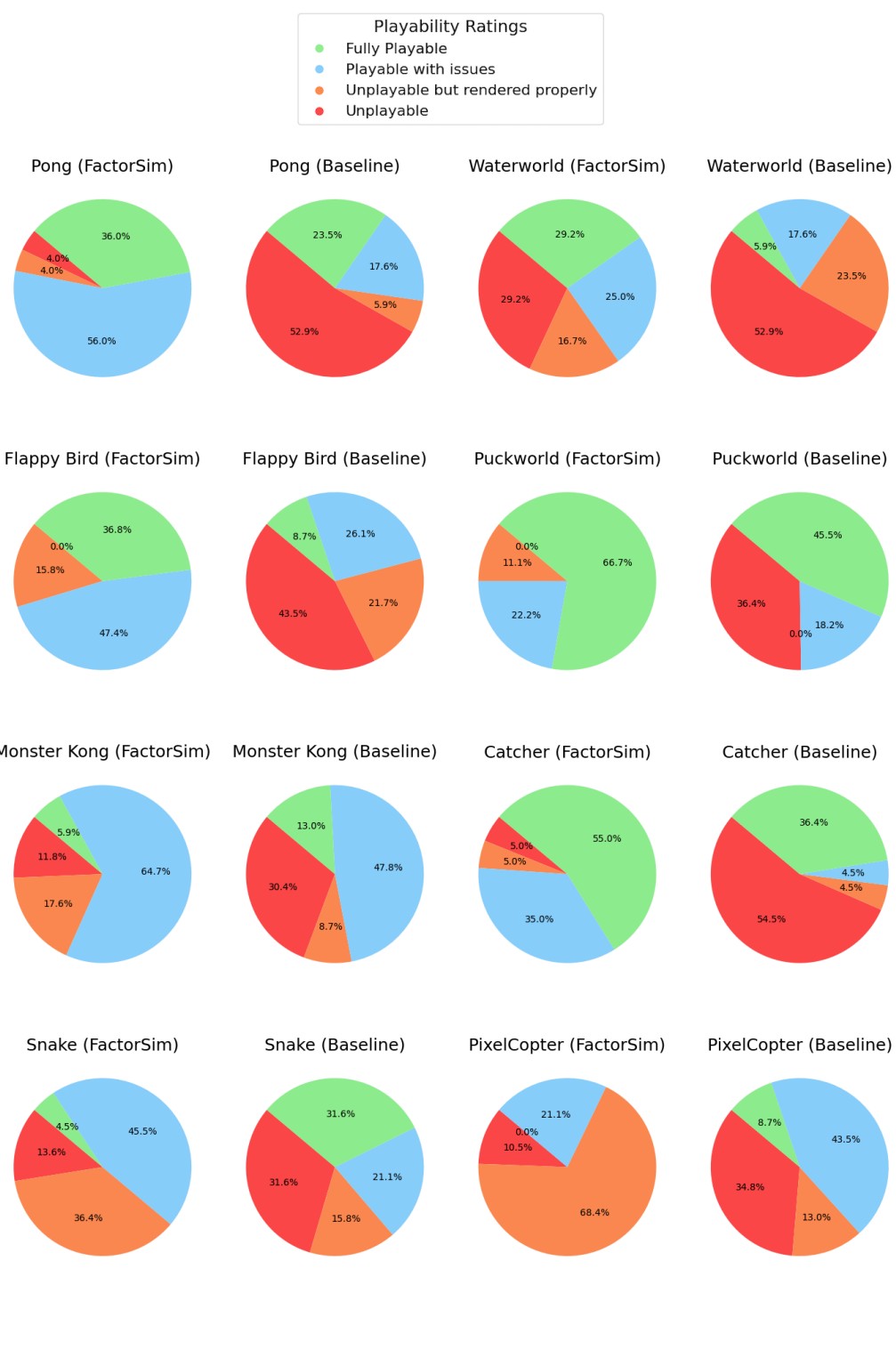

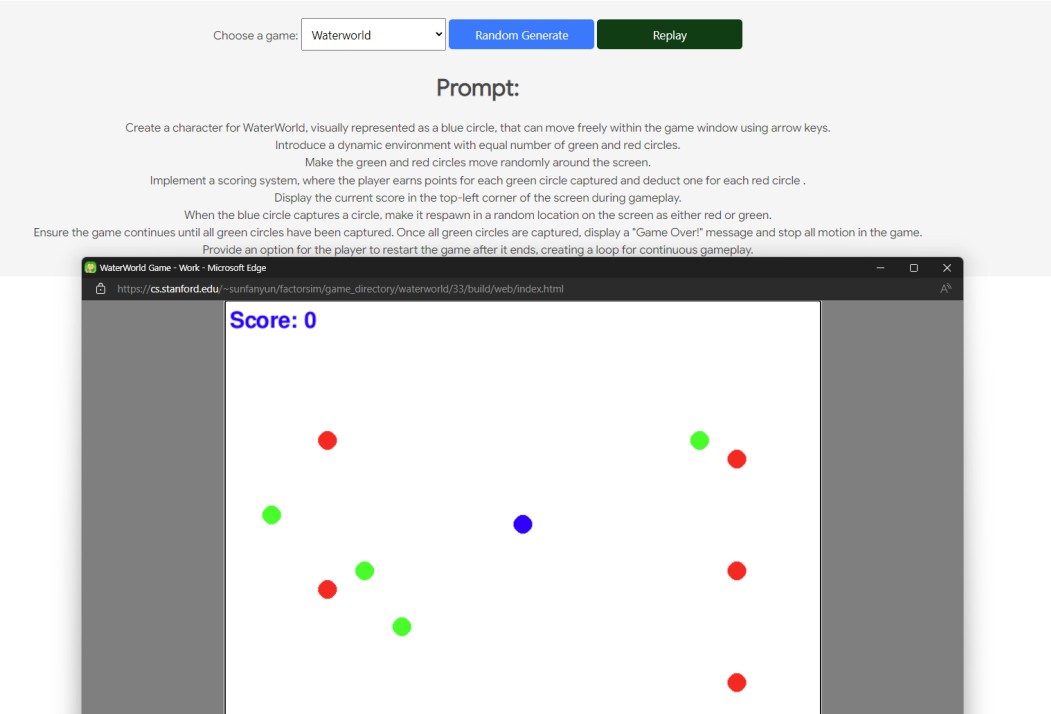

Figure 10: Human study interface screenshot.

