# OpenReview forum: "FactorSim: Generative Simulation via Factorized Representation"
_NeurIPS.cc/2024/Conference — NeurIPS 2024 poster_

### Official Review · Reviewer_6Px5 · 2024-07-06

**Soundness:** 3
**Presentation:** 3
**Contribution:** 2
**Rating:** 6
**Confidence:** 3

**Summary:**

This work presents FACTORSIM, a framework that converts any language specification into a complete simulation for training RL agents. FACTORSIM decomposes the input prompt into steps and uses a factored Partially Observable Markov Decision Process (POMDP) to minimize the context needed for each generation step. It also introduces a method to benchmark generative simulation and demonstrate the capability for FACTORSIM to be used in robotics setting.

**Strengths:**

1. Develop a robust pipeline for constructing game environments from language descriptions, which could significantly enhance the scalability of training generalist agents.
2. Formalize the framework as a Partially Observable Markov Decision Process (POMDP), reducing the need for full context during generation and improving outcomes.
3. Demonstrate the potential of this method to generalize to embodied scenarios.

**Weaknesses:**

1. The evidence for generalizing to embodied scenario is limited.
2. The successful rate in Table 1 and Figure 3 is low. Could there be some potential way to improve it?

**Questions:**

I have one primary concern: how could this be applied to the field of robotics and embodied AI?

**From this concern, several questions arise:**

- How is the background (or scenario) generated within this pipeline? In a 2D game setting, detailed descriptions generate the scenarios, but this can become extremely tedious as scenes become more complex, such as in an embodied environment. While language-guided scene generation could be a solution, how will it fit into the POMDP framework?
- The framework addresses three main components: controller, model, and view (rendering). In robotics, these aspects are typically handled by a physics simulation. How will this framework further contribute to the field of robotics? Currently, the paper shows potential for task generation in tabletop scenarios only.
- I am still unclear on how the Robotics Task Generation part is achieved by this pipeline.

**Some suggestions:**

- While it might be challenging to address this concern with experiments during the rebuttal period, more discussion on approaches and challenges would be beneficial.
- Any additional experiments that could demonstrate the pipeline's usage in robotics or embodied AI could help.

**Limitations:**

The limitations are addressed adequately.

---

> ### Author Rebuttal · Authors · 2024-08-07
>
> Thank you for your insightful feedback. We are glad that you share the excitement of the implication of this to the potential of scaling up the training of embodied generalist agents.
>
> # How can this be applied to the field of robotics and embodied AI?
> We present a novel method for factorizing simulation codebases, based on factorized POMDP representation, which allows for efficient context selection. Since most robotics tasks can be formulated as a POMDP, we believe that our method can beneficial in the field of robotics and embodied specifically in assembly related tasks at they require many dependent steps. To further demonstrate the superiority of our method, we also conduct additional experiments on 12 assembly-related tasks where FactorSim outperforms all Chain-of-Thought baselines used in GenSim by a large margin due to its ability to factorize the multi-step process into subtasks and generate these subtasks in separate steps, by using only the relevant context needed. Please find more details in the following section.
>
> # Details on the robotics experiment
>
> Refer to Figure 2 in the newly attached pdf for an overview of our robotics experiment. Below, we explain our experimental settings in more detail and how we apply FactorSim to generate robotics tasks.
>
> The paper GenSim [1] proposed a benchmark to investigate how well Large Language Models (LLMs) can design and implement simulation tasks and how to improve their performance on task generation. This benchmark includes a list of tasks described in the following format:
>
> - Task: Build-Car
> - task-name: build-car
> - task-description: Construct a simple car structure using blocks and cylinders.
> assets-used: [block/block.urdf, ball/ball-template.urdf]
>
> After the task code is generated, GenSim builds on top of CLIPort, evaluates the tasks by feeding them through a series of checkers that validate "syntax-correctness," "runtime-verification," and "task-completion."We followed the same experimental setting as described in their paper, except that we introduced an additional metric, "human-pass-rate" to measure for prompt alignment. This “human pass rate” checks whether the generated task adheres to the input prompt upon completion.
>
> FactorSim achieved state-of-the-art performance on their benchmark. To clarify how we apply FactorSim to the benchmark, we first describe how the POMDP is defined in this setting:
>
> - States: object states in the environment (color, object pose, size, scale)
> - Reward function: a set of functions that define how the rewards are given (e..g., a reward of X is given when the blue cube is in the bowl). This is analogous to the scoring functions are generated in the case of RL game generation.
> - Transitional function and Observation function: The underlying physics simulation and the robot assumed in CLIPort handle these aspects, so we don't generate them.
>
> We apologize if we didn’t make it clear how we applied FactorSim to generate robotic tasks. To summarize, a robotic task is no different than a RL game in that the final robotic task can be modeled as a POMDP. While the transition dynamics in this case are often handled by a physics simulator, we still need to generate the “reward function.” The only part left to generate is essentially the reward function, in the form of goal states. Objects in the scene and their arrangements are essentially state variables, and the functions are essentially functions that operate on these state variables to define whether a state should be given rewards. This is analogous to how in RL games we implement a reward function that takes in a set of state variables and updates them.
>
> # Additional robotics task generation experiment
> To further showcase FactorSim's effectiveness in generating robotics tasks, we form a list of 12 assembly tasks (i.e. build related tasks) and by combining the assembly tasks in GenSim's benchmark and by asking GPT-4 to generate additional assembly tasks such as "BuildLampPost" or "BuildDogHouse". As shown in Table 1 of the attached pdf to our general response, FactorSim outperforms all baselines by a large margin, including GenSim's two Chain-of-Thought baselines. We also provide visualizations of tasks that FactorSim is able to generate, demonstrating collection from using the oracle agent in the accompanying figure. These tasks use primitive assets like blocks and balls to form more complicated structures. We believe that scene generation fits into the POMDP framework, similar to what we achieved in our robotics task generation experiment.
>
> # Can we improve upon the results in Table 1 and Figure 3
> The low success rates testify to the difficulty of the tasks in the benchmark. We additionally ran AgentCoder [1], a SOTA code generation method that uses a multi-agent system to perform CoT prompting, on our benchmark. Please find the results in the General Response. While AgentCoder claims to refine code iteratively, it performs poorly because it relies on a test designer agent and a test executor agent to write quality test cases. However, due to the complexity of the tasks in our benchmark, the test designer agent tends to write incorrect or infeasible tests, leading to negative feedback. This points to FactorSim being an improvement over the standard "role-based" Chain of
> Thought decompositions.
>
> # How is the background generated?
> In Pygame learning environments, the game states are represented as a non-visual state, as exemplified by the documentation of the Catcher game: https://pygame-learning-environment.readthedocs.io/en/latest/user/games/catcher.html.
>
> [1] Huang, Dong, et al. "Agentcoder: Multi-agent-based code generation with iterative testing and optimisation." arXiv preprint arXiv:2312.13010 (2023).

---

> > ### Comment · Reviewer_6Px5 · 2024-08-12
> >
> > Thank you for the response. Most of my concerns have been addressed.
> >
> > While the main technical contribution appears to be in Code Synthesis, extending these skills to the field of robotics—a critical application area for code generation—also represents an important aspect of this paper. The authors have addressed this point well in their rebuttal.
> >
> > As a result, I am inclined to raise my assessment to a weak accept for this paper.
> >
> > BTW, it would be beneficial to improve the quality of the figures in the future. For instance, in Rebuttal Figure 2, the red dashed square intersects both the code and the outer square, which could be refined.

---

> ### Author Response · Authors · 2024-08-14
>
> Dear Reviewer 6Px5,
>
> Thank you for taking the time to read our rebuttal. We have refined our figures, as you suggested.
>
> Best regards,
>
> Authors.

---

### Official Review · Reviewer_yVST · 2024-07-11

**Soundness:** 2
**Presentation:** 3
**Contribution:** 2
**Rating:** 6
**Confidence:** 4

**Summary:**

The paper proposes a LLM prompting method to generate full game / robot simulations in code based on text descriptions.  Given a long text description, the method first utilizes an LLM to decompose it into multiple sentences, and then use them to iteratively generate and update simulation code. For each iteration, the code is generated and updated separately as three modules, i.e., controller, model and view. The update happens in a factorized way - the authors use the LLM to identify relevant state and context to avoid feeding the full generated code into LLM.

In experiments, the method is evaluated on game and robot simulation code generation benchmarks. The method shows superior results against other LLM prompting baselines in generating valid simulation code that aligns with text description.

**Strengths:**

- The proposed method exploits the structure of simulation to modularize and factorize code generation. This strategy significantly improves LLM's capability to generate full simulation code.
- The method is comprehensively evaluated on game and robot benchmarks.
- The paper is well written and easy to follow.

**Weaknesses:**

The major contribution of the paper seems to be a prompting technique crafted for the specific task of simulation code generation. While such a technique does improve performance on the task, my concern is it is neither fundamental nor sufficiently novel. The proposed prompting technique highlights two key designs:
- modularize simulation code generation manually, which aligns with the common practice to manually decompose a complex task into sub-tasks for LLMs to handle more effectively.
- extract relevant portion of code for LLM to consume and update, which is also an implementation-wise design that many works have already incorporated.

While the paper writes factorized POMDP formulations, they don't seem to make a difference on how the prompting method is implemented. So I'm concerned that the contribution of this paper is more as a practical application rather than a general method or framework.

**Questions:**

I'm curious what the failure modes of FactorSim is like.

**Limitations:**

See Weakness

---

> ### Author Rebuttal · Authors · 2024-08-07
>
> Thank you for dedicating your time to review our paper and for providing insightful feedback. We are glad to learn that you find our paper to be well-written and comprehensively evaluated.
>
> # Novelty
> We present a novel method for generating coded simulations that allows for efficient context selection; our method outperforms the Chain-of-Thought baseline in generating complex code, e.g., RL environments by a significant margin (See Table 1 in the main paper) and robotics tasks (See Figure 3 in the main paper and Table 1 in the pdf attached to our general response). More specifically, our method differs from the existing Chain-of-Thought methods by introducing a principled way to select context from the codebase, based on the factorized POMDP representation. It can be applied to any code generation task where the target generation can be modeled as a Partially Observable Markov Decision Process (POMDP), which is a flexible representation. The key advantage of our proposed representation is that it helps us model dependencies between different state variables and functions, instead of having to consider the entire codebase when making modifications to it or having to retrieve context code snippets based on some pre-defined similarity scores (e.g., akin to Retrieval Augmented Generation). To further showcase the novelty and effectiveness of FactorSim, we ran an additional SOTA baseline that uses a "role-based" chain of thought to perform code generation.
>
>
> | RL Env| GPT-4 w/ AgentCoder | GPT-4 w/ FactorSim |
> |----------------------|:----------------:|:----------------:|
> | Flappy Bird | 0.18 | 0.78 |
> | Catcher | 0.45 | 0.66 |
> | Snake | 0.27 | 0.44 |
> | Pixelcopter | 0.43 | 0.78 |
> | Pong | 0.43 | 0.61 |
> | Puckworld | 0.33 | 0.81 |
> | Waterworld | 0.20 | 0.62 |
> | Monster Kong | 0.23 | 0.44 |
>
> While AgentCoder claims to refine code iteratively, it performs poorly because it relies on a test designer agent and a test executor agent to write quality test cases. However, due to the complexity of the tasks in our benchmark, the test designer agent tends to write incorrect or infeasible tests, leading to negative feedback. This points to FactorSim being an improvement over the standard "role-based" Chain of Thought decompositions, and that it is non-trivial to generate simulations from complex textual specifications.
>
> # Failure modes and Limitations
> The improvement of FactorSim stems from better context provided in the prompts, allowing the LLMs to be more focused during code generation. Below, we discuss one failure mode and one limitation of FactorSim. The main failure mode we observe with FactorSim is when the context is selected incorrectly, leading to incorrect implementation. However, we find that the benefit of FactorSim outweighs the occasional errors LLMs make when selecting contexts, as shown in our experiments.
>
> In our experiment of generating robotics tasks, we find that all baselines, including our method, often ignore physical constraints necessary for the task to be completed by a robot. It is difficult for LLMs to consider context related to these "constraints" necessary for the task to be completed without being explicitly prompted. For example, if the LLM is prompted to generate the task "build a bridge", LLMs might generate a "bridge" block that is too small to span the two bottom items' distance. When the LLM is prompted to generate the task "put the ball in the container," the generated task might consist of a base container that is much smaller than the size of the ball. We leave this to future work.

---

> > ### Comment · Reviewer_yVST · 2024-08-10
> >
> > Thank you for the rebuttal! It's good to see how FactorSim performs against a recent baseline AgentCoder and what its failure modes are like. I thought about it carefully and agreed to the claimed novelty of the proposed method.
> >
> > I'm happy to increase the score to Weak Accept, given the clear writting and comprehensive evaluation of the paper.

---

> > > ### Author Response · Authors · 2024-08-14
> > >
> > > Dear Reviewer yVST,
> > >
> > > Thank you for taking the time to read our rebuttal!
> > >
> > > Best regards,
> > >
> > > Authors.

---

### Official Review · Reviewer_p3c7 · 2024-07-13

**Soundness:** 2
**Presentation:** 2
**Contribution:** 2
**Rating:** 4
**Confidence:** 3

**Summary:**

The paper proposed a factorized approach to generate simulated games via LLM code synthesis. The code idea is that one doesn't need to generate the entire code at once, but rather generate different part of a POMDP game, such as controller, model, and view. The generated simulation allows RL policies to train on top. The authors introduced a benchmark to evaluate the proposed framework and show good results in terms of prompt alignment, transfer ability and human evaluation.

**Strengths:**

The paper investigates an important problem, simulation generation. The evaluation over the mentioned environments is solid, spanning from automated tests to human evaluations.

**Weaknesses:**

1. The paper is poorly written. I have hands-on experience with almost all important concepts mentioned in the paper, yet still have a hard time understanding the paper, and have to read again and again including some code. Rather than talking about abstract terms like POMDP / factorization first, I think the authors can start easy with intuitions and explanations. The figure can also be improved. The main method figure shall spend more time showing what's special about "Factored POMDP" compared to prior methods. The benchmark claim should have its own section. The motivation is not clearly narrated either.  The world model section in related work doesn't seem to fit there.

2. On of the main contributions the authors listed in the introduction section is a benchmark. However, I think this benchmark seems to lack the technical depth I was expecting as a standalone contribution. I feel it's just a set of small metrics rather than benchmark.

3. The paper just lacks the level of technical contribution that meets my criteria for a Neurips paper. While there are many other prompting papers like CoT, ToT, the problem the paper is trying to solve is also very specific.

4. While I have experience with both LLM and robotics, I believe the authors should not put Robotics as primary area, but NLP or code synthesis community.

**Questions:**

In figures like figure 6, is the human pass rate based on the previous stage e.g. only executable code.

It seems that on open source model like llama 3, gensim with CoT is very close to factor sim. Can you explain the insights?

**Limitations:**

The author discussed the limitations of automated evaluation by conducting a human study. No outstanding negative societal impact.

---

> ### Author Rebuttal · Authors · 2024-08-07
>
> Thank you for your feedback! We are glad that you find this an important problem and that you find our evaluation solid and comprehensive.
>
> # Clarification of our Motivation and Novelty and why we chose Robotics as the primary area
> Recent advancements in foundational models have demonstrated their value in cognitive planning in robotics. The use of these models for obtaining lower-level policy is relatively underexplored, with many works focusing on direct control outputs [6, 7]. Our work leverages foundational models to generate simulations for training (i.e., Generative Simulation), a promising approach for training RL and robotics agents [1, 2]. However, LMs often fail at generating large amounts of complex code, leading to incorrect simulations and significant distributional shifts in downstream environments.
>
> By representing our codebase as a factorized POMDP, we can decompose the code into simpler chunks, reducing the need for extensive context. This method improves the accuracy of generating simulations with complex logic, outperforming all Chain-of-Thought-based baselines in generating prompt-aligned coded simulations. In zero-shot transfer experiments, more accurate code generation for simulations proves pivotal for the downstream training of RL and robotics agents. FactorSim’s zero-shot performance reaches 50% of direct training results in 5 out of 8 environments, whereas baselines show significant transfer in only 1-2 environments. Given that most robotics tasks can be formulated as a POMDP, our method could significantly impact the robotics community.
>
> # Presentation
> Thank you for your detailed feedback on our presentation. We understand the importance of legibility and have made the following improvements to our paper:
> - We included intuitive explanations for our method. We propose Factored POMDP representations to model an existing simulation codebase as a hypergraph, with nodes as state variables and hyperedges as functions relating to one or more state variables. Based on instructions for module additions, LLMs first select relevant state variables, then include only these variables and related functions in subsequent prompts.
> - We added a new figure to demonstrate how our formulation directly corresponds to prompt construction, with general explanations and concrete examples.
> - We included two new figures to explain our robotics task generation experiment and show visualizations of tasks that FactorSim successfully generates, which all other baselines fail to achieve.
> - We moved the world model part of the related section to the appendix.
>
> # Clarification on our Benchmark
> Pygame Learning Environment [3] is a Reinforcement Learning benchmark used in various existing works [4,5]. We extend these environments to include prompts with detailed specifications of game logic paired with system tests to test for “precise” prompt adherence ability in simulation code generation. When coupled with the original RL environments, our extension enables the evaluation of not only the accuracy of the generated code but also its usefulness in solving RL tasks (i.e., zero-shot transfer).
>
>
> # Additional experiments on the benchmark
> We additionally ran AgentCoder [8], a SOTA code generation method that uses a multi-agent system to perform CoT prompting, on our benchmark.
> | RL Env| GPT-4 w/ AgentCoder | GPT-4 w/ FactorSim |
> |----------------------|:----------------:|:----------------:|
> | Flappy Bird | 0.18 | 0.78 |
> | Catcher | 0.45 | 0.66 |
> | Snake | 0.27 | 0.44 |
> | Pixelcopter | 0.43 | 0.78 |
> | Pong | 0.43 | 0.61 |
> | Puckworld | 0.33 | 0.81 |
> | Waterworld | 0.20 | 0.62 |
> | Monster Kong | 0.23 | 0.44 |
>
> Additionally, we added two Atari games to our benchmark to increase the difficulty of our tasks. Here are provide some preliminary results:
>
> | RL Env | GPT-4 | GPT-4 w/ self debug | GPT-4 w/ Factor Sim |
> |----------------------|:----------------:|:----------------:|:----------------:|
> | Breakout | 0.13 | 0.10 | 0.40 |
> | Space Invaders | 0.10 | 0.14 | 0.28 |
>
> This benchmark allows us to investigate whether classical RL environments can be solved through the paradigm of generative simulation.
>
> # In figures like figure 6, is the human pass rate based on the previous stage e.g. only executable code.
>
> Yes. These four metrics (i.e. "syntax-correct", "runtime-verified", "task-completed", "human-pass") have an incremental structure from syntax to runtime to successfully generate demonstrations to human verification where failing the former metrics implies failing on the latter metrics.
>
> # Explain the insights re: open source models’ performance
> In our experiments, llama3 performs very well on generating specific tasks but exhibits instability, performing exceptionally on some tasks while poorly on others. Empirically, we observe that it resorts to "memorization" more, suggesting that if the memorized code is correct, it excels at generating those tasks.
> GenSim with CoT, both bottom-up and top-down, are competitive baselines that use a chain of 4-7 prompts. Despite this, FactorSim outperforms it, demonstrating its superior performance in task generation.
>
>
> [1] Xian, Zhou, et al. "Towards generalist robots: A promising paradigm via generative simulation."
>
> [2] Katara, Pushkal, Zhou Xian, and Katerina Fragkiadaki. "Gen2sim: Scaling up robot learning in simulation with generative models."
>
> [3] Tasfi, Norman. "Pygame learning environment." GitHub repository (2016).
>
> [4] Riemer, Matthew, et al. "Learning to learn without forgetting by maximizing transfer and minimizing interference." ICLR 2019.
>
> [5] Tucker, Aaron, Adam Gleave, and Stuart Russell. "Inverse reinforcement learning for video games."
>
> [6] Wang, Yen-Jen, et al. "Prompt a robot to walk with large language models."
>
> [7] Nasiriany, Soroush, et al. "Pivot: Iterative visual prompting elicits actionable knowledge for vlms."
>
> [8] Huang, Dong, et al. "Agentcoder: Multi-agent-based code generation with iterative testing and optimisation."

---

> ### Comment · Reviewer_p3c7 · 2024-08-09
>
> I acknowledge that I've read the rebuttal and other reviewers' opinions.
>
> Figure 1 in the rebuttal pdf definitely made the paper much more intuitive, allowing me to verify the original interpretation of the paper is correct. I recommend the authors to further improve its quality and put it in the paper whether it's accepted to Neurips or not.
>
> I still believe robotics should not be the primary area for reviewer allocation even given the author's response - just as reviewer EKr6 mentioned, the robot experiments seem like an afterthought. LLM / code synthesis shall be the better pool of reviewers, and I believe the AC shall consider this when making the final decision from our reviews.
>
> I am raising the score for the presentation & overall rating a bit since I find the general response/pdf helpful to understanding, yet still deem the paper unable to meet the acceptance threshold. I am willing to defend my rating, although other reviewers may have different opinions.

---

> > ### Author Response · Authors · 2024-08-11
> > **Thank you for reviewing our rebuttal**
> >
> > Thank you for taking the time to read our rebuttal and for your constructive feedback. We really appreciate your acknowledgment of the improvements we made.
> >
> > The subject area was indeed a challenging decision for us, and we would have chosen LLMs or Code Synthesis if they had been available. Looking back at our discussion before the submission, we first carefully listed down all available options:
> > - Machine vision
> > - Natural Language Processing
> > - Speech and Audio
> > - Deep Learning Architectures
> > - Generative Models
> > - Diffusion-based models
> > - Optimization for deep networks
> > - Evaluation (methodology, meta-studies, replicability, and validity)
> > - Online Learning
> > - Bandits
> > - Reinforcement Learning
> > - Active Learning
> > - Infrastructure (libraries, improved implementation and scalability, distributed solutions)
> > - Machine learning for healthcare
> > - Machine learning for physical sciences (for example: climate, physics)
> > - Machine learning for social sciences
> > - Machine learning for other sciences and fields
> > - Graph neural networks
> > - Neuroscience and cognitive science (neural coding, brain-computer interfaces)
> > - Optimization (convex and non-convex, discrete, stochastic, robust)
> > - Probabilistic methods (for example, variational inference, Gaussian processes)
> > - Casual Inference
> > - Robotics
> > - Interpretability and explainability
> > - Fairness
> > - Privacy
> > - Safety in machine learning
> > - Human-AI Interaction
> > - Learning theory
> > - Algorithmic game theory
> > - Other (please use sparingly, only use the keyword field for more details)
> >
> > We first narrowed it down to Generative Models, Reinforcement learning, Natural Language Processing, and Robotics. Then, we ruled out Generative Models as we are not creating a new generative model. Between Reinforcement Learning, Natural Language Process, and Robotics, we ultimately chose Robotics as the primary area for a couple of reasons: (a) we are not trying to solve a core NLP or RL problem, (b) the downstream application of our code generation method is RL and robotics, and (c) the message we hope to communicate to the community and the broader implications beyond our empirical results are more aligned with the robotics domain, as highlighted in our rebuttals.
> >
> > As you may see, none of these areas are a perfect match for our submission. We hope this clarifies the concern about our submission's subject area. Please don't hesitate to let us know if there are any other concerns that we may address to meet the acceptance threshold.

---

### Official Review · Reviewer_EKr6 · 2024-07-17

**Soundness:** 3
**Presentation:** 3
**Contribution:** 3
**Rating:** 6
**Confidence:** 3

**Summary:**

The paper introduces an LLM-based method for generting code for simulations. After generating the simulation of famous games based on their manual and description, the authors show that policies trained in these environments transfer well to the real games.

**Strengths:**

- **S.1 Great results.** I think that the results from Fig.3 are very impressive. Zero-shot transfer is very hard, and doing so much more reliably than vanilla GPT-4 is impressive.
- **S.2 Overall good idea.** The idea to deconstruct the game development task into M-V-C makes a lot of sense to me. I just thought that's already kind of captured in the POMDP formulation.
- **S.3 Good presentation.** The overall presentation and writing are good, although there is much left to be desired in terms of implementation details.

**Weaknesses:**

- **W.1 Implementation details and relationship to formulas.** I'm happy that there was code provided with the submissio and I hope that it will be released publicly because based purely on the main body of the paper, the method is not reproducible. Including the prompts in the appendix helps but I wish they were commented a bit more on why certain phrases and sections are there. And I'm wondering if the theory that's presented in the paper holds water wrt the actual instructions in the appendix. Because as far as I understand, there aren't any restrictions on what code the LLM can generate for each component, right? Also, the paper mentions graphs every so often and I don't know how they fit into this. I also think the context selection is crucial to your method and from the main body of the paper, it's completely unclear how that's implemented.
- **W.2 Missing examples.** Along a similar idea, I'd have loved some examples throughout the paper to illustrate what some of these instructions actually mean.
- **W.3 Unclear robotic experiments.** The robotic experiments seem to be more of an afterthought in the paper, and it's unclear what existing assets there are, what control code is assumed given, what camera parameters are assumed, etc.
- **W.4 Unclear input mapping.** The appendix mentions that the controller is given or that button presses always mean the same thing. I completely don't understand what's meant by that.

Overall, I think the paper shows a great idea and is probably beneficial to the community, but some work should go into tweaking the main body of the paper and making the method more clear and reproducible.

**Questions:**

I don't have any major questions wrt the work. There were a couple of points throughout the paper in the methods section, where I asked myself why this is relevant, but then this was cleared up a paragraph later.

**Limitations:**

The authors adequately address the limitations of their work.

---

> ### Author Rebuttal · Authors · 2024-08-07
>
> Thank you for taking the time to review our paper and for providing constructive feedback. We are glad you share our excitement about FactorSim's performance on the challenging task of zero-shot transfer. We appreciate your feedback and would like to address your questions.
>
> # Missing examples
> Thank you for your great suggestion! Following your advice, we have included two new figures in the pdf file attached to our general response. One is to illustrate FactorSim with a concrete example, and the other will explain our robotics experiment in more detail. See more detail below.
>
> # The connection between FactorSim and the formulations
> While LLMs are free to generate code, our formulation in the paper (i.e., Algorithm 1) directly informs what is included in the prompt as context. To clearly illustrate how FactorSim's prompts are constructed and map to the theory presented in Algorithm 1, we made a new figure and attached it to the PDF of our general response. FactorSim uses five prompts, as highlighted in grey boxes in the newly added figure. The first prompt corresponds to Equation 1 in our main paper. It decomposes the long and complex simulation specification into a list of steps to be implemented sequentially. The output from that is a list of instructions. For example, a step instruction could be "Introduce a blue agent rendered as a blue circle, allowing players to control it with arrow keys. The red puck should respawn when it collides with the blue agent." Then, at each step, FactorSim would add and modify the code of the existing codebase to incorporate the instructed change.
>
> During each step, FactorSim uses the second prompt (i.e. Context Selection Prompt, as shown in the figure) to select a set of state variables from the list of all state variables in the current codebase. This step corresponds to Equations 9 and 10 in Algorithm 1 in our main paper. The output of this prompt consists of a list of state variables, including state variables that are considered to be relevant to this module and new state variables that LLMs deem necessary to add to the system. After the list of context states is obtained, we will use it to retrieve the relevant functions in the codebase. A function is defined as relevant to a state variable if the state variable is used in the function. Our factorized POMDP representation essentially arranges a codebase into a (hyper-)graph where state variables are nodes and functions are hyperedges. After we obtain the list of relevant states and functions, they are used as context for the existing codebase in the following prompts the LLMs to generate the "model," "view," and "controller" part of the module. Note that not all steps require all three components; for instance, in our robotics experiment, only the "model" is generated as the "view" and "controller" are determined by the robot and the physics engine.
>
> # Clarification on our robotics experiment
>
> The paper GenSim proposed a benchmark to evaluate how well LLMs can design and implement simulation tasks, as well as improve their performance in task generation. We follow the same experimental setup described in their paper, with the addition of a new metric—"human pass rate"—to assess prompt alignment.
>
> In our experiments, we apply FactorSim to the benchmark proposed by the paper GenSim [2]. The benchmark, built on top of CLIPort [3], is designed to investigate how well Large Language Models (LLMs) can design and implement simulation tasks and how to improve their performance on task generation. This benchmark includes a list of tasks described in the following format:
>
> - task-name: build-car
> - task-description: Construct a simple car structure using blocks and cylinders.
> - assets-used: [block/block.urdf, ball/ball-template.urdf]
>
> The assets needed are provided in the prompts. The robot control and camera parameters are assumed to be given. Refer to Figure 2 in the attached PDF for an overview of our robotics experiment. We apply FactorSim to generate robotics tasks in a very similar way to how we generate RL games, as both can be modeled as POMDPs.
>
> To further demonstrate FactorSim's effectiveness in generating robotics tasks, we curate a list of 12 assembly-related tasks and show that FactorSim outperforms all Chain-of-Thought baselines used in GenSim by a large margin due to its ability to factorize the multi-step process into subtasks and generate these subtasks in separate steps, by using only the relevant context needed. Due to space limitations, please find more details in our response to reviewer 6Px5.
>
> # Unclear input mapping
> In our zero-shot transfer experiment, we train RL agents on generated code and test them on the original game implementation in the Pygame Learning Environment benchmark (PLE) [1]. In the Pygame Learning Environment (PLE), an action is defined as a "keyboard button." When we say, "The same button always means the same thing," we mean that a specific keyboard key consistently maps to the same in-game action during RL agent training. For example, pressing the spacebar in Flappy Bird will always trigger the flap action in training and testing environments. We will rewrite the sentence in our paper for better clarity. Thank you for pointing it out!
>
> # Reproducibility
> We are committed to releasing the code publicly. We will ensure that the code repository includes comprehensive documentation and examples to facilitate the reproduction of our results. We have sent our code, with instructions on how to run it, via an anonymous link to the AC, following the conference guidelines.
>
> ### References
>
> [1] Tasfi, Norman. "Pygame learning environment." GitHub repository (2016).
>
> [2] Wang, Lirui, et al. "Gensim: Generating robotic simulation tasks via large language models." ICLR 2024
>
> [3] Shridhar, Mohit, Lucas Manuelli, and Dieter Fox. "Cliport: What and where pathways for robotic manipulation." Conference on robot learning. PMLR, 2022.

---

### Author Rebuttal · Authors · 2024-08-07

We are grateful for the insightful feedback provided on our paper. We are encouraged to find that the reviewers found our paper well presented, comprehensively evaluated, achieved impressive zero-shot results, and shared our excitement about its applicability to training generalized agents. Below, we address some of the points raised by more than one reviewer and summarize the improvements we have made. We respond individually to points specific to each reviewer.

# Clarification of our contribution
With our goal of generating simulations that include long and complex prompt logic for downstream training in mind, our approach is to generate code for various simulation modules step-by-step, modifying and expanding the codebase. Every prompt has two components: the context (i.e., the existing codebase and how the new function will be used in conjunction with it) and the task specification (i.e., what to implement). Existing works showed that Retrieval-Augmented Generation can improve LLM’s performance, while irrelevant contexts can hurt [1,2]. FactorSim constructs the "context" portion of the prompts dynamically in a principled manner. While we agree that task decomposition or Chain-of-Thought (CoT) are proposals of existing work, our method for factorization and representation of dependencies for such code generation tasks are novel to the best of our knowledge.

To clarify our method, we provide a new figure in the pdf, illustrating how the formulation in Algorithm 1 directly corresponds to how the prompts are constructed. FactorSim consists of five prompts, highlighted in gray boxes. FactorSim first decomposes the long and complex prompt into a series of steps. Then, at each step, FactorSim modified the codebase according to the step instruction. If the codebase is written in an object-oriented paradigm, almost the entire codebase (i.e., 120 lines of code) would have to be included in the prompt as context since this logic pertains to many entities in the game. Instead, FactorSim maintains a factorized POMDP representation of the codebase, which is essentially a graph with state variables as nodes and functions as hyperedges:

| Name| Type|
|-|- |
|score| State Variable  |
| game_over| State Variable |
| green_puck_position| State Variable |
| red_puck_speed| State Variable|
| red_puck_radius| State Variable |
| blue_agent_position| State Variable  |
| …   | State Variable  |
| red_puck_respawn| Function |
| check_game_over_condition | Function|
| …| Function |

FactorSim uses PROMPT 2 to select a set of relevant state variables and then retrieve functions that pertain to the set of relevant context state variables according to the maintained graph structure. FactorSim constructs the “context” portion of the following code generation prompts with just the relevant state variables and the functions that modify one or more of the relevant state variables. In this example, instead of providing 120 lines of the codebase to the "context" part of the prompt, FactorSim only needs to include around 23 lines of code to the code generation prompts (i.e., prompt 3, 4, and 5 in Figure 1). Refer to Figure 1 for an overview of FactorSim, alongside an illustrative example.

# Broader Implications of our contribution
We believe in the significance of our work and FactorSim's broader implications. First, we demonstrate LLMs' ability to self-refine their prompt contexts for improved performance, unlike existing works that rely on similarity-based metrics to retrieve relevant code context [4].
Second, we believe the result of our zero-shot transfer experiment has important implications. Factorsim's zero-shot performance achieves 50% of the performance of training directly on the testing environments in 5 out of 8 Reinforcement Learning environments, compared to the best baseline's 1-2 out of 8. This suggests the importance of automated task generation and its potential for scaling up the training of generalist embodied agents.

# Improvements we have made:
1. **New Figure**: include a new figure to explain FactorSim, with concrete examples, and how our formulation directly corresponds to how the prompts are constructed. Refer to Figure 1 in the pdf.

2. **Additional code generation experiments**: We include a comparison between FactorSim and the SOTA code generation method (i.e., AgentCoder [3]). More detail on the experimental result can be found in our response to reviewer p3c7.

|Env|GPT-4 w/ AgentCoder| GPT-4 w/ FactorSim|
|-|:-:|:-:|
|Flappy Bird|0.18|0.78
|Catcher|0.45|0.66|
|Snake|0.27|0.44|
|Pixelcopter|0.43 | 0.78 |
|Pong | 0.43 | 0.61 |
|Puckworld | 0.33 | 0.81 |
|Waterworld | 0.20 | 0.62 |
|Monster Kong | 0.23 | 0.44 |


3. **Details on our robotics experiments**: We include a new figure that overviews how our robotics experiments are conducted and how FactorSim is utilized. Refer to our response to reviewer 6Px5.

4. **Additional robotics task generation experiments**: refer to Table 1 in the pdf and our response to reviewer 6Px5.

5. **Benchmark Extension**: We have extended our benchmark to include two classical Atari RL environments (i.e., breakout and space_invaders), which are included in the code we submitted. Results can be found in our response to reviewer p3c7.

6. **Reproducibility** To mitigate concerns regarding reproducibility, we have also sent an anonymous link to our codebase with instructions on how to run our code to the AC in an official comment, following the conference guidelines.

References:

[1] Shi, Freda, et al. "Large language models can be easily distracted by irrelevant context."

[2] Levy, Mosh, Alon Jacoby, and Yoav Goldberg. "Same task, more tokens: the impact of input length on the reasoning performance of large language models."

[3] Huang, Dong, et al. "Agentcoder: Multi-agent-based code generation with iterative testing and optimisation."

[4] Zhang, Fengji, et al. "Repocoder: Repository-level code completion through iterative retrieval and generation."

---

### Decision · Program_Chairs · 2024-09-25

**Decision:**

Accept (poster)

**Comment:**

This paper introduces FACTORSIM, an LLM-based method for generating simulation code from language input. The core concept is to iteratively generate and update different parts of a POMDP game, such as the controller, model, and view.

**strengths**

* Importance of the problem that this paper tackles and interesting idea.

* Extensive evaluation and impressive results including zero-shot transfer.

**weaknesses/suggestions**

* Clarification on robotics applications: Although the original explanation of the robotic experiments was unclear, the authors have addressed the concerns raised by several reviewers.

I think the paper makes a nice contribution that the community will find valuable. However, I encourage the authors to think carefully about how to reflect the comments or resolve the questions from reviewers in the camera-ready version.